# SyncTwin: Transparent Treatment Effect Estimation under Temporal Confounding

## Abstract

Estimating causal treatment effects using observational data is a problem with few solutions when the confounder has a temporal structure, e.g. the history of disease progression might impact both treatment decisions and clinical outcomes. For such a challenging problem, it is desirable for the method to be transparent — the ability to pinpoint a small subset of data points that contribute most to the estimate and to clearly indicate whether the estimate is reliable or not. This paper develops a new method, SyncTwin, to overcome temporal confounding in a transparent way. SyncTwin estimates the treatment effect of a target individual by comparing the outcome with its synthetic twin, which is constructed to closely match the target in the representation of the temporal confounders. SyncTwin achieves transparency by enforcing the synthetic twin to only depend on the weighted combination of few other individuals in the dataset. Moreover, the quality of the synthetic twin can be assessed by a performance metric, which also indicates the reliability of the estimated treatment effect. Experiments demonstrate that SyncTwin outperforms the benchmarks in clinical observational studies while still being transparent.

## 1 Introduction

Estimating the causal individual treatment effect (ITE) on patient outcomes using observational data (observational studies) has become a promising alternative to clinical trials as large-scale electronic health records become increasingly available (Booth & Tannock, 2014). Figure 1 illustrates a common setting in medicine and it will be the focus of this work (DiPietro, 2010): an individual may start the treatment at some observed time (black dashed line) and we want to estimate the ITE on the outcomes *over time* after the treatment starts (shaded area). The key limitation of observational studies is that treatment allocation is not randomised but typically influenced by prior measurable static covariates (e.g. gender, ethnicity) and temporal covariates (e.g. all historical medical diagnosis and conditions, squares in Figure 1). When the covariates also modulate the patient outcomes, they lead to the *confounding bias* in the direct estimation of the ITE (Psaty et al., 1999).

Although a plethora of methods overcome the confounding bias by adjusting for the *static* covariates (Yoon et al., 2018; Yao et al., 2018; Louizos et al., 2017; Shalit et al., 2017; Li & Fu, 2017; Alaa & van der Schaar, 2017; Johansson et al., 2016), few existing works take advantage of the temporal covariates that are measured irregularly over time (Figure 1) (Bica et al., 2020; Lim et al., 2018; Schulam & Saria, 2017; Roy et al., 2017). Overcoming the confounding bias due to temporal covariates is especially important for medical research as clinical treatment decisions are often based on the temporal progression of a disease. Transparency is highly desirable in such a challenging problem.

Figure 1: Illustration of a treated individual. Yellow dots represent the outcomes under no treatment.

Although transparency is a general concept, we will focus on two specific aspects (Arrieta et al., 2020). (1) **Explainability**: the method should estimate the ITE of any given individual (the *target individual*) based on a small subset of other individuals (*contributors*) whose amount of contribution can be quantified (e.g using a weight between 0 and 1). Although the estimate of different target individuals may depend on different contributors, the method can always shortlist the few contributors for the expert to understand the

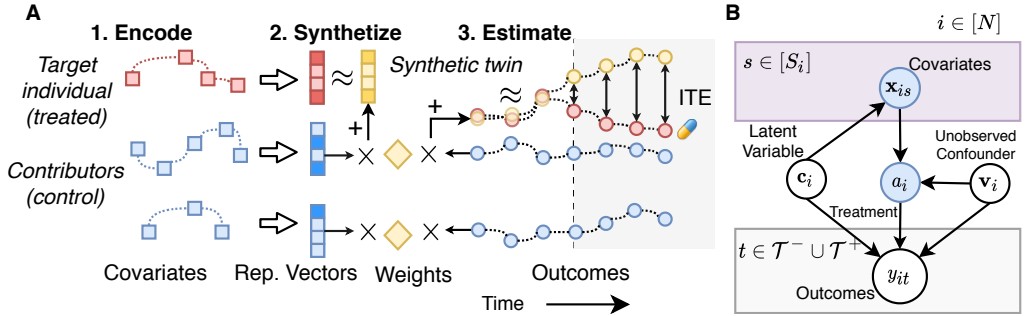

Figure 2: **A**: Illustration of SyncTwin (shaded area: the time points after the treatment starts). 1. Temporal covariates are encoded as representation vectors. 2. The synthetic twin of a treated target individual is constructed as the weighted average of the few contributors from the control group. 3. The difference between the observed outcome and the synthetic twin outcome estimates ITE. **B**: the DAG of the data generating model (Sec. 2).

rationale for each estimate. (2) **Trustworthiness**: the method should identify the target individuals whose ITE cannot be reliably estimated due to violation of assumptions, lack of data, or other failure modes. Being transparent about what the method *cannot* do improves the overall trustworthiness because it guides the experts to only use the method when it is deemed reliable.

Inspired by the well-established Synthetic Control method in Statistics and Econometrics (Abadie et al., 2010; Abadie, 2019), we propose SyncTwin, a transparent ITE estimation method which deals with temporal confounding. Figure 2 A illustrates the schematics of SyncTwin. SyncTwin starts by encoding the irregularly-measured temporal covariates as representation vectors. For each treated target individual, SyncTwin selects and weights few contributors from the control group based on their representation vectors and the sparsity constraint. SyncTwin proceeds to construct a synthetic twin whose representation vector and outcomes are the weighted average of the contributors. Finally, the ITE is estimated as the difference between the target individual's and the Synthetic Control's outcomes *after* treatment. The difference in their outcomes *before* treatment indicates the quality of the synthetic twin and whether the model assumptions hold. If the target individual and synthetic twin do not match in pre-treatment outcomes, the estimated ITE should not be considered trustworthy.

**Transparency of SyncTwin**. SyncTwin achieves *explainability* by selecting only a few contributors for each target individual. It achieves *trustworthiness* because it quantifies the confidence one should put into the estimated ITE as the difference between the target and synthetic pre-treatment outcomes.

## 2 PROBLEM SETTING

We consider a clinical observational study with $N$ individuals indexed by $i \in [N] = \{1, \ldots, N\}$. Let $a_i \in \{0, 1\}$ be the treatment indicator with $a_i = 1$ if $i$ started to receive the treatment at some time and $a_i = 0$ if $i$ never initiated the treatment. We realign the time steps such that all treatments were initiated at time $t = 0$. Let $\mathcal{I}_1 = \{i \in [N] \mid a_i = 1\}$ and $\mathcal{I}_0 = \{i \in [N] \mid a_i = 0\}$ be the set of the treated and the control respectively. Denote $N_0 = |\mathcal{I}_0|$ and $N_1 = |\mathcal{I}_1|$ as the sizes of the groups. The time $t = 0$ is of special significance because it marks the initiation of the treatment (black dashed line in Figure 1). We call the period $t < 0$ the pre-treatment period and the period $t \geq 0$ the treatment period (shaded area in Figure 1).

Temporal covariates are observed during the pre-treatment period only and may influence the treatment decision and the outcome. Let $\mathbf{X}_i = [\mathbf{x}_{is}]_{s \in [S_i]}$ be the sequence of covariates $\mathbf{x}_{is} \in \mathbb{R}^D$, which includes $S_i \in \mathbb{N}$ observations taken at times $t \in \mathcal{T}_i = \{t_{is}\}_{s \in [S_i]}$, where all $t_{is} \in \mathbb{R}$ and $t_{is} < 0$. Note that $\mathbf{x}_{is}$ may also include static covariates whose values are constant over time. To allow the covariates to be sampled at different frequencies, let $\mathbf{m}_{is} \in \{0, 1\}^D$ be the masking vector with $m_{isd} = 1$ indicating the $d$th element in $\mathbf{x}_{is}$ is sampled.

The outcome of interest is observed both before and after the treatment. In many cases, the researchers are interested in the outcomes measured at regular time intervals (e.g. the monthly average blood pressure). Hence, let $\mathcal{T}^- = \{-M, \ldots, -1\}$ and $\mathcal{T}^+ = \{0, \ldots, H - 1\}$ be the observation times

before and after treatment initiation. In this work, we focus on real-valued outcomes $y_{it} \in \mathbb{R}$ observed at $t \in \mathcal{T}^- \cup \mathcal{T}^+$. We arrange the outcomes after treatment into a $H$-dimensional vector denoted as $\mathbf{y}_i = [y_{it}]_{t \in \mathcal{T}^+} \in \mathbb{R}^H$. Similarly define pre-treatment outcome vector $\mathbf{y}_i^- = [y_{it}]_{t \in \mathcal{T}^-} \in \mathbb{R}^M$.

Using the potential outcome framework (Rubin, 2005), let $y_{it}(a_i) \in \mathbb{R}$ denote the potential outcome at time $t$ in a world where $i$ received the treatment as indicated by $a_i$. Let $\mathbf{y}_i(1) = [y_{it}(1)]_{t \in \mathcal{T}^+} \in \mathbb{R}^H$, and $\mathbf{y}_i^-(1) = [y_{it}(1)]_{t \in \mathcal{T}^-} \in \mathbb{R}^M$, similarly for $\mathbf{y}_i(0)$ and $\mathbf{y}_i^-(0)$. The individual treatment effect (ITE) is defined as $\tau_i = \mathbf{y}_i(1) - \mathbf{y}_i(0) \in \mathbb{R}^H$. Under the consistency assumption (discussed later in details), the *factual* outcome is observed $\mathbf{y}_i(a_i) = \mathbf{y}_i$, which means for any $i \in [N]$ only the unobserved *counterfactual* outcome $\mathbf{y}_i(1 - a_i)$ needs to be estimated in order to estimate the ITE. To simplify the notations, we focus on estimating the ITE for the treated, i.e. $\hat{\tau}_i = \mathbf{y}_i(1) - \hat{\mathbf{y}}_i(0)$ for $i \in \mathcal{I}_1$, though the same approach applies to the control $i \in \mathcal{I}_0$ and new units $i \notin [N]$ without loss of generality (A.5).

SyncTwin relies on the following assumptions. (1) *Consistency*, also known as Stable Unit Treatment Value Assumption (Rubin, 1980): $y_{it}(a_i) = y_{it}, \forall i \in [N], t \in \mathcal{T}^- \cup \mathcal{T}^+$. (2) *No anticipation*, also known as causal systems (Abbring & Van den Berg, 2003; Dash, 2005): $y_{it} = y_{it}(1) = y_{it}(0)$, $\forall t \in \mathcal{T}^-, i \in [N]$. (3) *Data generating model*: the assumed directed acyclic graph is visualized in Figure 2 B (Pearl, 2009), where we introduce two variables $c_i \in \mathbb{R}^K$ and $v_i \in \mathbb{R}^U$ in addition to the previously defined ones. The latent variable $c_i$ is the common cause of $y_{it}(0)$ and $x_{is}$, and it indirectly influences $a_i$ through $x_{is}$. As we show later, SyncTwin tries to learn and construct a synthetic twin that has the same $c_i$ as the target. The variable $v_i$ is an *unobserved* confounder. Although SyncTwin, like all other ITE methods, works better without unobserved confounders (i.e. $v_i = 0, \forall i \in [N]$), we develop a unique checking procedure in Equation (4) to validate if there exists $v_i \neq 0$. We also demonstrate that under certain favourable conditions, SyncTwin can overcome the impact of the $v_i$. To establish the theoretical results, we further assume $y_{it}(0)$ follows a latent factor model with $c_i, v_i$ as the latent "factors"(Bai & Ng, 2008):

$$y_{it}(0) = \mathbf{q}_t^\top \mathbf{c}_i + \mathbf{u}_t^\top \mathbf{v}_i + \xi_{it}, \quad \forall t \in \mathcal{T}^- \cup \mathcal{T}^+, \tag{1}$$

where $\mathbf{q}_t \in \mathbb{R}^K, \mathbf{u}_t \in \mathbb{R}^U$ are weight vectors and $\xi_{it}$ is the white noise. We require the weight vectors to have $||\mathbf{q}_t|| = 1, \forall t \in \mathcal{T}^- \cup \mathcal{T}^+$ (Xu, 2017), which does not reduce the expressiveness of the model. We further require the dimensionality of the latent factor to be smaller than the number of time steps before or after treatment, i.e. $K < \min(M, H)$. Furthermore, let $\mathbf{Q}^- = [\mathbf{q}_t]_{t \in \mathcal{T}^-} \in \mathbb{R}^{M \times K}$ and $\mathbf{Q} = [\mathbf{q}_t]_{t \in \mathcal{T}^+} \in \mathbb{R}^{H \times K}$ denote the matrices that stack all the weight vectors $\mathbf{q}$'s before and after treatment as rows respectively. The latent factor model assumption may seem restrictive but as we show in Appendix A.4 it is applicable to many scenarios. In the simulation study (5.1) we further show SyncTwin performs well even when the data is not generated using model (1) but instead using a set of differential equations. We compare our assumptions with those used in the related works in Appendix A.3.

# 3 RELATED WORK

## 3.1 SYNTHETIC CONTROL

Similar to SyncTwin, Synthetic control (SC) (Abadie, 2019) and its extensions (Athey et al., 2018; Amjad et al., 2018) estimate ITE based on Synthetic Control outcomes. However, when applied to temporal confounding, SC will *flatten* the temporal covariates $[x_{is}]_{s \in [S_i]}$ into a fixed-sized (high-dimensional) vector $\underline{x}_i$ and use it to construct the twin. As a result, SC does not allow the covariates to be variable-length or sampled at different frequencies (otherwise $\underline{x}_i$'s dimensionality will vary across individuals). In contrast, SyncTwin can gracefully handle these irregularities because it constructs the twin using the representation vectors. Moreover, the covariates $\underline{x}_i$ may contain observation noise and other sources of randomness that do not relate to the outcome or the treatment. Enforcing the target and the twin to have similar $\underline{x}_i$ will inject these irrelevant noise to the twin, a situation we call *over-match* (because it resembles over-fit). Over-match undermines ITE estimation as we show in the simulation study in Section 5.1. Finally, SC assumes $y_{it}(0) = \mathbf{q}_t^\top \underline{x}_i + \mathbf{u}_t^\top \mathbf{v}_i + \xi_{it}$, i.e. the flattened covariates $\underline{x}_i$ linearly predicts $y_{it}(0)$, which is a special case of our assumption (1) and unlikely to hold for many medical applications.

### 3.2 COVARIATE ADJUSTMENT WITH DEEP LEARNING

In the *static* setting, the covariate adjustment methods fit two functions (deep neural networks) to predict the outcomes with and without treatment i.e. $\hat{\mathbf{y}}_i(0) = f_0(\mathbf{x}_i)$ and $\hat{\mathbf{y}}_i(1) = f_1(\mathbf{x}_i)$ (Johansson et al., 2016; Shalit et al., 2017; Yao et al., 2018; Yoon et al., 2018). The ITE is then estimated as $\hat{\tau}_i = f_1(\mathbf{x}_i) - f_0(\mathbf{x}_i)$. Under this framework, various methods have been proposed to address *temporal confounding* (Lim et al., 2018; Bica et al., 2020). However, these methods generally lack transparency because the black-box neural networks cannot easily pinpoint the contributors for each prediction. Moreover, the prediction accuracy before treatment cannot directly measure the confidence or trustworthiness for the predictions after treatment because the network is very nonlinear and non-stationary. Lastly, Bica et al. (2020) and Lim et al. (2018) are applicable to a more general setting where the treatment can be turned on and off over time whereas SyncTwin assumes the outcomes will continue to be influenced by the treatment after the treatment starts.

**Works with similar terminology**. Several works in the literature use similar terms such as "twin" while most of them are not related to SyncTwin. We discuss these works in Appendix A.6.

## 4 TRANSPARENT ITE ESTIMATION VIA SYNCTWIN

To explain when and why SyncTwin gives a valid ITE estimate, let us assume that we have learned a representation $\tilde{\mathbf{c}}_i$ that approximates the latent variable $\mathbf{c}_i$, $\forall i \in [N]$ in Equation 1. For a target individual $i \in \mathcal{I}_1$, let $\boldsymbol{b}_i = [b_{ij}]_{j \in \mathcal{I}_0} \in \mathbb{R}^{N_0}$ be a vector of weights, each associated with a control individual. A synthetic twin can be generated using $\boldsymbol{b}_i$ as

$$\hat{\mathbf{c}}_i = \sum_{j \in \mathcal{I}_0} b_{ij}\tilde{\mathbf{c}}_j, \qquad \hat{\mathbf{y}}_{it}(0) = \sum_{j \in \mathcal{I}_0} b_{ij}\mathbf{y}_{jt}(0) = \sum_{j \in \mathcal{I}_0} b_{ij}\mathbf{y}_{jt}, \ \ \forall t \in \mathcal{T}^- \cup \mathcal{T}^+, \qquad (2)$$

where $\hat{\mathbf{c}}_i$ is the synthetic representation and $\hat{\mathbf{y}}_{it}(0)$ is the synthetic outcome under no treatment. The last equality follows from the consistency assumption. Let $\hat{\mathbf{y}}_i(0) = [\hat{\mathbf{y}}_{it}(0)]_{t \in \mathcal{T}^+}$ be the post-treatment synthetic outcome vector, and similarly $\hat{\mathbf{y}}_i^- = [\hat{\mathbf{y}}_{it}(0)]_{t \in \mathcal{T}^-}$. The ITE of $i$ can be estimated as

$$\hat{\tau}_i = \mathbf{y}_i(1) - \hat{\mathbf{y}}_i(0) = \mathbf{y}_i - \sum_{j \in \mathcal{I}_0} b_{ij}\mathbf{y}_j, \qquad (3)$$

where again the last equality follows from the consistency assumption. We should highlight that $\mathbf{y}_i$ and $\mathbf{y}_j$, $\forall j \in \mathcal{I}_0$ in the equation above are the *observed* outcomes. Hence, $\mathbf{b}_i$ is the only free parameter that influences the ITE estimator $\hat{\tau}_i$. The following two distances are central to the training and inference procedure:

$$\mathrm{d}_i^c = \|\hat{\mathbf{c}}_i - \tilde{\mathbf{c}}_i\|, \qquad \mathrm{d}_i^y = \|\hat{\mathbf{y}}_i^- - \mathbf{y}_i^-\|_1, \qquad (4)$$

where $\|\cdot\|$ is the vector $\ell_2$-norm and $\|\cdot\|_1$ is the vector $\ell_1$-norm.

**Minimizing $\mathrm{d}_i^c$ to construct synthetic twins**. $\mathrm{d}_i^c$ indicates how well the synthetic twin matches the target individual in representations. Intuitively, we should seek to construct a twin who closely matches the target by minimizing $\mathrm{d}_i^c$. This intuition is verified in Proposition 1 (proved in A.1.1).

**Proposition 1** (Bias bound on ITE with no unobserved confounders). Suppose that $\mathbf{v}_i = \mathbf{0}$, $\forall i \in [N]$ and $\mathrm{d}_i^c = 0$ for some $i \in \mathcal{I}_1$ ($\mathbf{v}_i$ and $\mathrm{d}_i^c$ are defined in Equation 1 and 4 respectively), the absolute value of the expected difference in the true and estimated ITE of $i$ is bounded by:

$$|\mathbb{E}[\hat{\tau}_i] - \mathbb{E}[\tau_i]| \le |\mathcal{T}^+| \|\sum_{j \in \mathcal{I}_0} b_{ij}\mathbf{c}_j - \mathbf{c}_i\| \le |\mathcal{T}^+| \Big( \sum_{j \in \mathcal{I}_0} \|\mathbf{c}_j - \tilde{\mathbf{c}}_j\| + \|\mathbf{c}_i - \tilde{\mathbf{c}}_i\| \Big). \qquad (5)$$

Here we show that when $\mathrm{d}_i^c$ is minimized at zero and there is no unobserved confounder, the bias on the ITE estimate only depends on how close the learned representation $\tilde{\mathbf{c}}$ is to the true latent variable $\mathbf{c}$. We will use use representation learning to uncover the latent variable $\mathbf{c}$ in the next section.

**Using $\mathrm{d}_i^y$ to measure trustworthiness**. By definition, $\mathrm{d}_i^y$ indicates how well the synthetic pre-treatment outcomes match the target individual's outcomes. Intuitively, matching the outcomes *before* treatment is a prerequisite for a good estimate of the ITE *after* treatment (Equation 2 and 3). We formalize this intuition in Proposition 2, which is proved in Appendix A.1.1.

**Proposition 2** (Trustworthiness of SyncTwin under no hidden confounders). Suppose that all the outcomes are generated by the model in Equation 1 with the unobserved confounders equal to zero s.t. $\mathbf{v}_i = \mathbf{0}, \forall i \in [N]$, and that we reject the estimate $\hat{\tau}_i$ if the pre-treatment error $\mathrm{d}_i^y$ on $\mathcal{T}^-$ is larger than $\delta |\mathcal{T}^-|/|\mathcal{T}^+|$, the post-treatment ITE estimation error on $\mathcal{T}^+$ is below $\delta$.

Here we show that if we would like to ensure the ITE error to fall below a pre-specified threshold $\delta$, we should reject the estimate $\hat{\tau}_i$ when the distance $\mathrm{d}_i^y > \delta |\mathcal{T}^-|/|\mathcal{T}^+|$ assuming no unobserved confounder. In other words, $\mathrm{d}_i^y$ can be used as an evaluation metric to access whether the estimated ITE is trustworthy.

**Situation with unobserved confounders.** In presence of the unobserved confounders $\mathbf{v}_i \neq \mathbf{0}$, SyncTwin cannot guarantee to correctly estimate the ITE. However, $\mathrm{d}_i^y$ can still indicate whether $\mathbf{v}_i$ has a significant impact on the pre-treatment outcomes, i.e. the unobserved confounders may *exist* but only *weakly* influence the outcomes before treatment. We discuss unobserved confounders in detail in Appendix A.1.2.

## 4.1 LEARNING TO REPRESENT TEMPORAL COVARIATES

In this section, we show how SyncTwin learns the representation $\tilde{\mathbf{c}}_i$ as a proxy for the latent variable $\mathbf{c}_i$ using a sequence-to-sequence architecture as depicted in Figure 3 (A) and discussed below.

**Architecture.** SyncTwin is agnostic to the exact choice of architecture as long as the network translates the covariates into a single representation vector (encode) and reconstructs the covariates from that representation (decode). For this reason, we use the well-proven sequence-to-sequence architecture (Seq2Seq) (Sutskever et al., 2014) with an encoder similar to the one proposed in Bahdanau et al. (2015) and a standard LSTM decoder (Hochreiter & Schmidhuber, 1997).

The encoder first obtains a sequence of representations at each time step using a recurrent neural network. Instead of using the bi-directional LSTM as in Bahdanau et al. (2015), we use a GRU-D network because it is designed to encode irregularlly-sampled temporal observations (Che et al., 2018). This gives us the sequence $\mathbf{h}_{is} = \text{GRU-D}(\mathbf{h}_{i,s-1}, \mathbf{x}_{is}, \mathbf{m}_{is}, t_{is}), \forall s \in [S_i]$. Since our goal is to obtain a single representation vector rather than a sequence of representations, we aggregate the sequence of $\mathbf{h}_{is}$ using the same attentive pooling method as in Bahdanau et al. (2015). The final representation vector $\tilde{\mathbf{c}}_i$ is obtained as: $\tilde{\mathbf{c}}_i = \sum_{s \in [S_i]} \alpha_{is} \mathbf{h}_{is}$, where $\alpha_{is} = \mathbf{r}^\top \mathbf{h}_{is}/\sqrt{K}$ is the attention weight and $\mathbf{r} \in \mathbb{R}^K$ is the attention parameter (Vaswani et al., 2017).

The decoder uses the representation $\tilde{\mathbf{c}}_i$ to reconstruct $\mathbf{x}_{is}$ at time $t_{is} \forall s \in [S_i]$. Since the timing information $t_{is}$ may be lost in $\tilde{\mathbf{c}}_i$ due to aggregation, we reintroduce it to the decoder by first obtaining a sequence of time representations $\mathbf{o}_{is} = \mathbf{k}_0 + \mathbf{w}_0^\top t_{is}$, where $\mathbf{o}_{is}, \mathbf{k}_0, \mathbf{w}_0 \in \mathbb{R}^K$, and then concatenating each with $\tilde{\mathbf{c}}_i$ to obtain: $\mathbf{e}_{is} = \tilde{\mathbf{c}}_i \oplus \mathbf{o}_{is} \in \mathbb{R}^{2K}$. Reintroducing timing information during decoding is a standard practice in Seq2Seq models for irregular time-series (Rubanova et al., 2019; Li & Marlin, 2020). Furthermore, using time representation $\mathbf{o}_{is}$ instead of time values $t_{is}$ is inspired by the success of positional encoding in the self-attention architecture (Vaswani et al., 2017; Gehring et al., 2017). The decoder then applies a LSTM autoregressively on the time-aware representations $\mathbf{e}_{is}$ to decode $\mathbf{g}_{is} = \text{LSTM}(\mathbf{g}_{i,s-1}, \mathbf{e}_{is}), \forall s \in [S_i]$, where $\mathbf{g}_{is} \in \mathbb{R}^K$. Finally, it uses a linear layer to obtain the reconstructions: $\hat{\mathbf{x}}_{is} = \mathbf{k}_1 + \mathbf{W}_1 \mathbf{g}_{is}$, where $\mathbf{k}_1 \in \mathbb{R}^D, \mathbf{W}_1 \in \mathbb{R}^{D \times K}$.

**Loss functions.** We train the networks with the weighted sum of the supervised loss $\mathcal{L}_s$ and the reconstruction loss $\mathcal{L}_r$ (Figure 3 A):

$$\mathcal{L}_s(\mathcal{D}_0) = \sum_{i \in \mathcal{D}_0} ||\tilde{\mathbf{Q}} \cdot \tilde{\mathbf{c}}_i - \mathbf{y}_i(0)||, \qquad \mathcal{L}_r(\mathcal{D}_0, \mathcal{D}_1) = \sum_{i \in \mathcal{D}_0 \cup \mathcal{D}_1} \sum_{s \in [S_i]} ||(\tilde{\mathbf{x}}_{is} - \mathbf{x}_{is}) \odot \mathbf{m}_{is}||, \quad (6)$$

where $\mathcal{D}_0 \subseteq \mathcal{I}_0, \mathcal{D}_1 \subseteq \mathcal{I}_1$, $\mathbf{m}_{is}$ is the masking vector (Section 2), $\odot$ represents element-wise product and $\tilde{\mathbf{Q}} \in \mathbb{R}^{H \times K}$ is a trainable parameter and $|| \cdot ||$ is the $L_2$ norm. Intuitively, the supervised loss $\mathcal{L}_s$ ensures that the learned representation $\tilde{\mathbf{c}}_i$ to be a *linear* predictor of the outcomes under no treatment $\mathbf{y}_i(0)$. Here a linear function $\tilde{\mathbf{y}}_i(0) := \tilde{\mathbf{Q}} \cdot \tilde{\mathbf{c}}_i$ is used to be consistent with the data generating model (1). Using a nonlinear function here might lead to smaller $\mathcal{L}_s$, but it will not uncover the latent variable $\mathbf{c}_i$ as desired. We justify the supervised loss in Proposition 3 below and present the proof and detailed discussions in Appendix A.1.1.

**Proposition 3** (Error bound on the learned representations). Suppose that $\mathbf{v}_i = \mathbf{0}, \forall i \in [N]$ ($\mathbf{v}_i$ is defined in Equation 1), the total error on the learned representations for the control, i.e., the first term

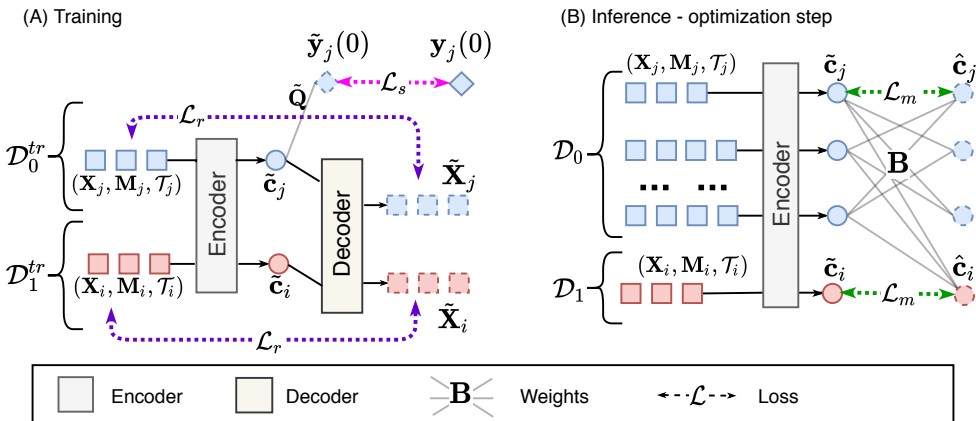

Figure 3: Illustration of the loss functions. (A) The representation networks are trained using $\mathcal{L}_s$ and $\mathcal{L}_r$ in Equation 6. Note that the supervised loss $\mathcal{L}_s$ only applies to the control. (B) Validation and inference involve optimizing the matching loss $\mathcal{L}_m$ in Equation 8. Note the encoder needs to be fixed during optimization.

in the upper bound of the absolute value of the expected difference in the true and estimated ITE (R.H.S of Equation 5), is bounded as follows:

$$\sum_{j \in \mathcal{I}_0} \|\mathbf{c}_j - \tilde{\mathbf{c}}_j\| \leq \beta \mathcal{L}_s + \sum_{j \in \mathcal{I}_0} \|\xi_j\|, \tag{7}$$

where $\mathcal{L}_s$ is the supervised loss in Equation 6 and $\xi_j$ is the white noise in Equation 1.

## 4.2 Constructing synthetic twins

**Constraints**. We require the weights $\mathbf{b}_i$ in Equation 2 to satisfy two constraints (1) positivity: $b_{ij} \geq 0 \,\forall i \in [N]$, $j \in \mathcal{I}_0$, and (2) sum-to-one: $\sum_{j \in \mathcal{I}_0} b_{ij} = 1, \forall i \in [N]$. The constraints are needed for three reasons. (1) The constraints reduce the solution space of $\mathbf{b}_i$ and serve as a *regularizer*. Regularizing is vital because the dimensionality of $\mathbf{b}_i \in \mathbb{R}^{N_0}$ can easily exceed ten thousand in observational studies. (2) The constraints encourage the solution to be sparse by fixing the $\ell_1$-norm of $\mathbf{b}_i$ to be one i.e. $||\mathbf{b}_i||_1 = 1$ (Tibshirani, 1996). Better sparsity leads to fewer contributors and better transparency. (3) Finally, the constraints ensure that the synthetic twin in Equation 2 is the weighted average of the contributors. Therefore the weight $b_{ij}$ directly translates into the "contribution" or "importance" of $j$ to $i$, further improving the transparency.

**Matching loss**. The matching loss finds weight $\mathbf{b}_i$ so that the synthetic twin and the target individual match in representations, as depicted in Figure 3 (B).

$$\mathcal{L}_m(\mathcal{D}_0, \mathcal{D}_1) = \sum_{i \in \mathcal{D}_1} ||\tilde{\mathbf{c}}_i - \sum_{j \in \mathcal{D}_0} b_{ij} \tilde{\mathbf{c}}_j||_2^2, \tag{8}$$

where again $\mathcal{D}_0 \subseteq \mathcal{I}_0$ and $\mathcal{D}_1 \subseteq \mathcal{I}_1$. We use the Gumbel-Softmax reparameterization detailed in Appendix A.9 to optimize $\mathcal{L}_m$ under the constraints (Jang et al., 2016; Maddison et al., 2016).

## 4.3 Training, Validation and Inference

As is standard in machine learning, we perform model training, validation and inference (testing) on three disjoint datasets. On a high level, we train the encoder and decoder on the training data using the loss functions described in Section 4.1. The validation data is then used to validate and tune the hyper-parameters of the encoder and decoder. Finally, we fix the encoder and optimize the matching loss $\mathcal{L}_m$ on the testing data to find the weight $\mathbf{b}_i$, which leads to the ITE estimate using Equation 3. The detailed procedure is described in A.8. The hyperparamter sensitivity is studied in A.13.

## 5 EXPERIMENTS

### 5.1 SIMULATION STUDY

In this simulation study, we evaluate SyncTwin on the task of estimating the LDL cholesterol-lowering effect of statins, a common drug prescribed to hypercholesterolaemic patients. We simulate the ground truth ITE using the widely adopted Pharmacokinetic-Pharmacodynamic model in the literature (Faltaos et al., 2006; Yokote et al., 2008; Kim et al., 2011).

$$\frac{d\mathrm{p}_t}{dt} = \mathrm{k}_t^{in} - k \cdot \mathrm{p}_t; \quad \frac{d\mathrm{d}_t}{dt} = \mathrm{a}_t - h \cdot \mathrm{d}_t; \quad \frac{d\mathrm{y}_t}{dt} = k \cdot \mathrm{p}_t - \frac{\mathrm{d}_t}{\mathrm{d}_t + d_{50}} k \cdot \mathrm{y}_t. \quad (9)$$

where $\mathrm{y}_t$ is the LDL cholesterol level (outcome) and $\mathrm{a}_t$ is the indicator of statins treatment. The interpretation of all other variables involved are presented in Appendix A.10.

**Data generation**. Following our convention, the individuals are enrolled at $t = 0$, the covariates are observed in $\mathcal{T} = [-S, 0)$, where $S \in \{15, 25, 45\}$, and the ITE is to be estimated in the period $\mathcal{T}^+ = [0, 4]$. We start by generating $\mathrm{k}_t^{in}$ for each individual from the following mixture distribution:

$$\mathrm{k}_{it}^{in} = \mathbf{g}_i^\top \boldsymbol{f}_t; \quad \mathbf{g}_i = \delta_i \mathbf{e}_{i1} + (1 - \delta_i)\mathbf{e}_{i2}; \quad \delta_i \overset{\text{iid}}{\sim} \text{Bern}(p); \quad \mathbf{e}_{in} \overset{\text{iid}}{\sim} \text{N}(\mu_n, \Sigma_n), \ n = 1, 2 \quad (10)$$

where $\boldsymbol{f}_t \in \mathbb{R}^6$ are the Chebyshev polynomials, $\text{Bern}(p)$ is the Bernoulli distribution with success probability $p$ and $\text{N}(\mu_n, \Sigma_n)$ is the Gaussian distribution. To introduce confounding, we vary $p$ for the treated and the control: $p = p_0, \forall i \in \mathcal{I}_0$ and $p = 1, \forall i \in \mathcal{I}_1$, where $p_0$ controls the degree of confounding bias. After that, the variables $\mathrm{p}_t, \mathrm{d}_t, \mathrm{y}_t$ are obtained by solving Equation 9 using `scipy` (Virtanen et al., 2020) and adding independent white noise $\epsilon \sim \text{N}(0, 0.1)$ to the solution. The temporal variables defined above give us the covariates $\mathbf{x}_t = \{\mathrm{k}_t^{in}, \mathrm{y}_t, \mathrm{p}_t, \mathrm{d}_t\}$. Finally, we introduce irregular sampling by creating masks $\mathbf{m}_{it} \sim \text{Bern}(m)$, where probability $m \in \{0.3, 0.5, 0.7, 1\}$.

**Benchmarks**. From the Synthetic Control literature, we considered the original Synthetic Control method (SC) (Abadie et al., 2010), Robust Synthetic Control (RSC) (Amjad et al., 2018) and MC-NNM (Athey et al., 2018). From the deep learning literature, we compared against Counterfactual Recurrent Network (CRN) (Bica et al., 2020) and Recurrent Marginal Structural Network (RMSN) (Lim et al., 2018), which are the state-of-the-art methods to estimate ITE under temporal confounding. In addition, we included a modified version of the CFRNet, which was originally developed for the static setting (Shalit et al., 2017). To allow the CFRNet to handle temporal co-

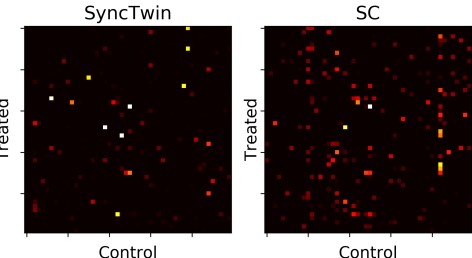

Figure 4: Heatmap of the weights $\mathbf{b}_i$ learned by SyncTwin and SC. Each row represents one $\mathbf{b}_i$.

variates, we replaced its fully-connected encoder with the encoder architecture used by SyncTwin (Section 4.1). We also included the counterfactual Gaussian Process (CGP) (Schulam & Saria, 2017) and One-nearest Neighbour Matching (1NN) (Stuart, 2010) as baselines. The implementation details of all benchmarks are available in Appendix A.7. We also included two ablated versions of SyncTwin. SyncTwin-$\mathcal{L}_r$ is trained only with reconstruction loss and SyncTwin-$\mathcal{L}_s$ only with supervised loss.

**Main results**. We evaluate the mean absolute error (MAE) on ITE estimation: $\frac{1}{N_1} \sum_{i=1}^{N_1} ||\tau_i - \hat{\tau}_i||_1$. In table 6 the parameter $p_0$ controls the level of confounding bias (smaller $p_0$, larger bias). Additional results for different sequence length $S$ and sampling irregularity $m$ are shown in Appendix A.11. SyncTwin achieves the best or equally-best performance in all cases. The full SyncTwin with both loss functions also consistently outperforms the versions trained only with $\mathcal{L}_r$ or $\mathcal{L}_s$. As discussed in Section 4 (and Appendix A.1.1), training with only reconstruction loss $\mathcal{L}_r$ leads to significant performance degradation. It is worth highlighting that the data generating model used in this simulation (9) is not the same as SyncTwin's assumed latent factor model (1). This implies that SyncTwin may still achieve good performance when the assumed model (1) does not *exactly* hold.

SC, RSC and MC-NNM underperform because their assumption that the flattened covariates $\underline{\mathbf{x}}_i$ *linearly* predict the outcome is violated (Section 3). Furthermore, Table 2 shows the synthetic twin created by SC matches the target covariates $\underline{\mathbf{x}}_i$ consistently better than SyncTwin, yet produces worse ITE estimates. This suggests that matching covariates better may not lead to better ITE estimate

Table 1: Mean absolute error on ITE under different levels of confounding bias $p_0$. $m = 1$ and $S = 25$ are used. Estimated standard deviations are shown in the parentheses. The best performer is in bold.

| Method | $N_0 = 200$ | | | $N_0 = 1000$ | | |
|---|---|---|---|---|---|---|
| | $p_0 = 0.1$ | $p_0 = 0.25$ | $p_0 = 0.5$ | $p_0 = 0.1$ | $p_0 = 0.25$ | $p_0 = 0.5$ |
| SyncTwin-Full | **0.324 (.038)** | **0.144 (.012)** | **0.119 (.008)** | **0.141 (.012)** | **0.106 (.006)** | **0.093 (.005)** |
| SyncTwin-$\mathcal{L}_r$ | 0.353 (.039) | 0.170 (.015) | 0.139 (.010) | 0.256 (.026) | 0.145 (.012) | 0.101 (.006) |
| SyncTwin-$\mathcal{L}_s$ | 0.336 (.039) | 0.170 (.015) | 0.120 (.008) | 0.144 (.012) | 0.113 (.007) | 0.127 (.010) |
| SC | 0.340 (.041) | 0.151 (.024) | 0.149 (.018) | 0.258 (.050) | 0.166 (.034) | 0.214 (.036) |
| RSC | 0.837 (.044) | 0.360 (.020) | 0.321 (.018) | 0.310 (.016) | 0.298 (.014) | 0.302 (.014) |
| MC-NNM | 1.160 (.059) | 0.612 (.031) | 0.226 (.011) | 0.527 (.029) | 0.159 (.008) | 0.124 (.006) |
| CFRNet | 0.895 (.077) | 0.411 (.037) | 0.130 (.007) | 0.411 (.038) | 0.175 (.013) | 0.106 (.007) |
| CRN | 1.045 (.064) | 0.546 (.039) | 0.360 (.024) | 0.864 (.052) | 0.767 (.040) | 0.357 (.021) |
| RMSN | 0.390 (.031) | 0.362 (.028) | 0.332 (.026) | 0.447 (.041) | 0.386 (.034) | 0.385 (.032) |
| CGP | 0.660 (.043) | 0.610 (.039) | 0.561 (.035) | 0.826 (.056) | 0.693 (.047) | 0.602 (.038) |
| 1NN | 1.866 (.099) | 1.721 (.091) | 1.614 (.078) | 2.446 (.131) | 1.746 (.106) | 1.384 (.083) |

Table 2: Mean absolute error between the observed covariates $\underline{\mathbf{x}}_i$ and synthetic twin's covariates $\hat{\underline{\mathbf{x}}}_i$. SC matches the covariates better yet produces worse ITE estimate (Table 1), suggesting it is over-matching. The average distance between any two individuals is 0.95, much larger than all values reported in the table.

| Method | $N_0 = 200$ | | | $N_0 = 1000$ | | |
|---|---|---|---|---|---|---|
| | $p_0 = 0.1$ | $p_0 = 0.25$ | $p_0 = 0.5$ | $p_0 = 0.1$ | $p_0 = 0.25$ | $p_0 = 0.5$ |
| SyncTwin-Full | 0.343 (.029) | 0.203 (.014) | 0.179 (.011) | 0.469 (.037) | 0.223 (.015) | 0.175 (.012) |
| SyncTwin-$\mathcal{L}_r$ | 0.321 (.028) | 0.192 (.015) | 0.182 (.015) | 0.250 (.019) | 0.190 (.013) | 0.195 (.013) |
| SC | **0.236 (.027)** | **0.117 (.014)** | **0.111 (.011)** | **0.155 (.025)** | **0.110 (.019)** | **0.128 (.020)** |

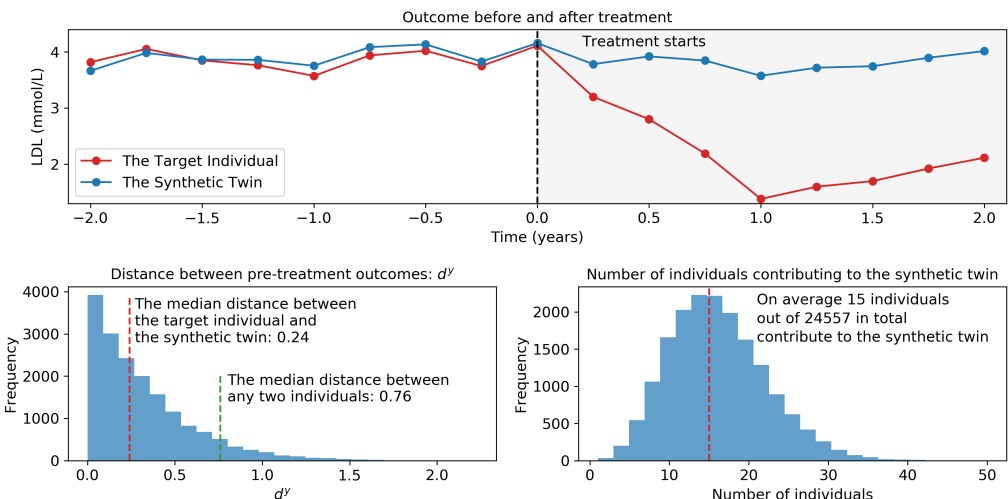

Figure 5: Illustration of the transparency of SyncTwin. Top: the outcomes (LDL) before and after treatment of a target individual and its synthetic twin. Bottom left: histogram of distance $\mathrm{d}^y$ (Equation 4). Bottom right: histogram of number of contributors used to construct the synthetic twin.

because the covariates are noisy and the method might over-match (Section 3). In addition, Figure 4 visualizes the weights $\mathbf{b}_i$ of SyncTwin and SC in a heatmap. We can clearly see that SyncTwin produces sparser weights because SC needs to use more contributors to construct the twin that (over-)matches $\underline{\mathbf{x}}_i$. Quantitative evaluation of the sparsity is provided in Appendix A.12. These findings verify our belief that constructing twins in the representation space (SyncTwin) rather than in the high-dimensional observation space (SC) leads to better performance and transparency.

## 5.2 EXPERIMENT ON REAL DATA

**Purpose of study**. We present an clinical observational study using SyncTwin to estimate the LDL Cholesterol-lowering effect of statins in the first year after treatment (Ridker & Cook, 2013).

**Data Source**. We used medical records from English National Health Service general practices that contributed anonymised primary care electronic health records to the Clinical Practice Research Datalink (CPRD), covering approximately 6.9 percent of the UK population (Herrett et al., 2015). CPRD was linked to secondary care admissions from Hospital Episode Statistics, and national mortality records from the Office for National Statistics. We defined treatment initiation as the date of first CPRD prescription and the outcome of interest was measured LDL cholesterol (LDL). Known risk factors for LDL were selected as temporal covariates measured before treatment initiation: HDL Cholesterol, Systolic Blood Pressure, Diastolic Blood Pressure, Body Mass Index, Pulse, Creatinine, Triglycerides and smoking status. Our analysis is based on a subset of 125,784 individuals (Appendix A.15) which was split into three equally-sized subsets for training, validation and inference, each with 17,371 treated and 24,557 controls.

**Evaluation**. We evaluate our models using the average treatment effect on the treated group (ie, ATT $= E(\tau_i | a_i = 1)$) to directly correspond to the reported treatment effect in randomised clinical trials, e.g. The Heart Protection Study reported an a change of -1.26 mmol/L (SD=0.06) in LDL cholesterol for participants randomised to statins versus placebo (Group et al., 2007; 2002). We use the sample average on the testing set to estimate the ATT as $\sum_{i \in \mathcal{D}_1^{te}} \hat{\tau}_{it} / |\mathcal{D}_1^{te}|$, where $\mathcal{D}_1^{te}$ are the individuals in the testing set who received the treatment. SyncTwin estimates the ATT to be **-1.25** mmol/L (SD 0.01), which is very close to the results from the clinical trial. In comparison, CRN and RMSN estimate the ATT to be **-0.72** mmol/L (SD 0.01) and **-0.83** mmol/L (SD 0.01) respectively. Other benchmark methods either cannot handle irregularly-measured covariates or do not scale to the size of the dataset. Our result suggests SyncTwin is able to overcome the confounding bias in the complex real-world datasets.

**Transparent ITE estimation**. For each individual, we can visualize the outcomes before and after the treatment and compare them with the synthetic twin in order to sense-check the estimate. The individual shown in Figure 5 (top) has a sensible ITE estimate because the synthetic twin matches its pre-treatment outcomes closely over time. In addition to visualization, we can calculate the distance $d^y$ (Equation 4) to quantify the difference between the pre-treament outcomes. From Figure 5 (bottom left) we can see in most cases the distance is small with a median of 0.24 mmol/L (compared to the population average distance 0.76 mmol/L). This means if the expert can only tolerate an error of 0.24 mmol/L on ITE estimation, half of the estimates (those with $d^y \leq 0.24$ mmol/L) can be accepted (Section 4). The estimates are also explainable due to the sparsity of SyncTwin. As shown in Figure 5 (bottom right) on average only 15 (out of 24,557) individuals contribute to the synthetic twin.

## 6 CONCLUSION

In this work, we present SyncTwin, an transparent ITE estimation method that deals with temporal confounding and has a broad range of applications in clinical observational studies and beyond. Combining the Synthetic Control method and deep representation learning, SyncTwin achieves transparency and strong performance in both simulated and real data experiments.

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

# A   APPENDIX

## A.1   THEORETICAL RESULTS

### A.1.1   SITUATION WITH NO UNOBSERVED CONFOUNDERS

**Proposition 1** Bias bound on ITE with no unobserved confounders. Suppose that $\mathbf{v}_i = \mathbf{0}, \forall i \in [N]$ and $\mathrm{d}_i^c = 0$ for some $i \in \mathcal{I}_1$ ($\mathbf{v}_i$ and $\mathrm{d}_i^c$ are defined in Equation 1 and 4 respectively), the absolute value of the expected difference in the true and estimated ITE of $i$ is bounded by:

$$|\mathbb{E}[\hat{\tau}_i] - \mathbb{E}[\tau_i]| \leq |\mathcal{T}^+| \| \sum_{j \in \mathcal{I}_0} b_{ij} \mathbf{c}_j - \mathbf{c}_i \| \leq |\mathcal{T}^+| \Big( \sum_{j \in \mathcal{I}_0} \|\mathbf{c}_j - \tilde{\mathbf{c}}_j\| + \|\mathbf{c}_i - \tilde{\mathbf{c}}_i\| \Big). \tag{11}$$

*Proof.* We start the proof by observing

$$
\begin{aligned}
|\mathbb{E}[\hat{\tau}_i] - \mathbb{E}[\tau_i]| &= \sum_{t \in \mathcal{T}^+} |\mathbb{E}[\hat{\mathbf{y}}_{it}(0)] - \mathbb{E}[\mathbf{y}_{it}(0)]| \\
&= \sum_{t \in \mathcal{T}^+} |\mathbf{q}_t^\top (\sum_{j \in \mathcal{I}_0} b_{ij} \mathbf{c}_j - \mathbf{c}_i)| \\
&\leq \sum_{t \in \mathcal{T}^+} \|\mathbf{q}_t\| \cdot \| \sum_{j \in \mathcal{I}_0} b_{ij} \mathbf{c}_j - \mathbf{c}_i \| \\
&= |\mathcal{T}^+| \| \sum_{j \in \mathcal{I}_0} b_{ij} \mathbf{c}_j - \mathbf{c}_i \|
\end{aligned}
\tag{12}
$$

where the first equation follows from the definition of ITE in Section 4. The second equation follows from Equation 1 and 2 together with the fact that $\mathbf{v}_i = \mathbf{0}, \forall i \in [N]$. The third line follows from Cauchy–Schwarz inequality. The fourth line uses the fact that $\|\mathbf{q}_t\| = 1$. By definition, $\mathrm{d}_i^c = 0$ implies $\sum b_{ij} \tilde{\mathbf{c}}_j = \tilde{\mathbf{c}}_i$. Continuing the proof,

$$
\begin{aligned}
\| \sum_{j \in \mathcal{I}_0} b_{ij} \mathbf{c}_j - \mathbf{c}_i \| &= \| \sum_{j \in \mathcal{I}_0} b_{ij} (\mathbf{c}_j - \tilde{\mathbf{c}}_j) - (\mathbf{c}_i - \tilde{\mathbf{c}}_i) \| \\
&\leq \sum_{j \in \mathcal{I}_0} b_{ij} \|\mathbf{c}_j - \tilde{\mathbf{c}}_j\| + \|\mathbf{c}_i - \tilde{\mathbf{c}}_i\| \\
&\leq \sum_{j \in \mathcal{I}_0} \|\mathbf{c}_j - \tilde{\mathbf{c}}_j\| + \|\mathbf{c}_i - \tilde{\mathbf{c}}_i\|,
\end{aligned}
\tag{13}
$$

where the second line follows from the triangular inequality and the third line relies on $\sum_{j \in \mathcal{I}_0} b_{ij} = 1$ and $b_{ij} \geq 0, \forall j \in \mathcal{I}_0$. Combining inequality 12 and 13, we prove the inequalities in Equation 11.   $\square$

**Justification for the matching loss and $\mathrm{d}_i^c$.** Proposition 1 presents a justification for minimizing $\mathrm{d}_i^c$ (or the matching loss $\mathcal{L}_m$). Essentially, when the synthetic representations are matched with the target ($\mathrm{d}_i^c = 0$), the bias in ITE estimate is controlled by how close the learned representations $\tilde{\mathbf{c}}$ is to the true latent variable $\mathbf{c}$ up to an arbitrary linear transformation $\Lambda$. An important implication is that the learned representation $\tilde{\mathbf{c}}$ does not need to be equal to $\mathbf{c}$, instead the learning algorithm only needs to identify $\mathbf{c}$ up to a linear transformation. Of course, Proposition 1 also implies that $|\mathbb{E}[\hat{\tau}_i] - \mathbb{E}[\tau_i]| \leq \sum_{j \in \mathcal{I}_0} \|\mathbf{c}_j - \tilde{\mathbf{c}}_j\| + \|\mathbf{c}_i - \tilde{\mathbf{c}}_i\|$ when $\Lambda$ is taken to be the identity matrix instead of the minimizer.

**Proposition 2** Trustworthiness of SyncTwin under no hidden confounders. Suppose that all the outcomes are generated by the model in Equation 1 with the unobserved confounders equal to zero s.t. $\mathbf{v}_i = \mathbf{0}, \forall i \in [N]$, and that we reject the estimate $\hat{\tau}_i$ if the pre-treatment error $\mathrm{d}_i^y$ on $\mathcal{T}^-$ is larger than $\delta |\mathcal{T}^-|/|\mathcal{T}^+|$, the post-treatment ITE estimation error on $\mathcal{T}^+$ is below $\delta$.

*Proof.* As a reminder, $\mathbf{Q}^- = [\mathbf{q}_t]_{t \in \mathcal{T}^-}$ and $\mathbf{Q} = [\mathbf{q}_t]_{t \in \mathcal{T}^+}$ denote the matrix that stacks all the weight vectors $\mathbf{q}$'s before and after treatment as rows respectively where each $\mathbf{q}_t$ satisfies that $\|\mathbf{q}_t\| = 1$ in

Equation 1. The error $d_i^y$ in Equation 4 can be decomposed into a representation error and a white noise error,

$$
\begin{aligned}
d_i^y &= \|\hat{\mathbf{y}}_i^- - \mathbf{y}_i^-\|_1 \\
&= \|\sum_{j \in \mathcal{I}_0} b_{ij} \mathbf{y}_j^- - \mathbf{y}_i^-\|_1 \\
&= \|\sum_{j \in \mathcal{I}_0} b_{ij} (\mathbf{Q}^- \mathbf{c}_j + \xi_j) - (\mathbf{Q}^- \mathbf{c}_i + \xi_i)\|_1 \\
&= \|\mathbf{Q}^- \big(\sum_{j \in \mathcal{I}_0} b_{ij} \mathbf{c}_j - \mathbf{c}_i\big)\|_1 + \|\sum_{j \in \mathcal{I}_0} b_{ij} \xi_j - \xi_i\big)\|_1 \\
&\leq \sum_{t \in \mathcal{T}^-} \|\mathbf{q}_t\| \|\sum_{j \in \mathcal{I}_0} b_{ij} \mathbf{c}_j - \mathbf{c}_i\| + \|\sum_{j \in \mathcal{I}_0} b_{ij} \xi_j - \xi_i\|_1 \\
&\leq |\mathcal{T}^-| \|\sum_{j \in \mathcal{I}_0} b_{ij} \mathbf{c}_j - \mathbf{c}_i\| + \|\sum_{j \in \mathcal{I}_0} b_{ij} \xi_j - \xi_i\|
\end{aligned}
\tag{14}
$$

We can not estimate the error from the representation and white noise on the last line of Equation 14. Conservatively, we can say the representation error itself is larger or equal to $d_i^y$ such that

$$
|\mathcal{T}^-| \|\sum_{j \in \mathcal{I}_0} b_{ij} \mathbf{c}_j - \mathbf{c}_i\| \geq d_i^y,
$$

i.e.,

$$
\|\sum_{j \in \mathcal{I}_0} b_{ij} \mathbf{c}_j - \mathbf{c}_i\| \geq d_i^y / |\mathcal{T}^-|.
\tag{15}
$$

The post-treatment error is upper bounded as follows,

$$
\begin{aligned}
|\mathbb{E}[\hat{\tau}_i] - \mathbb{E}[\tau_i]| &= |\mathbb{E}[\hat{\mathbf{y}}_{it}(0)] - \mathbb{E}[\mathbf{y}_{it}(0)]| \\
&= \sum_{t \in \mathcal{T}^+} |\mathbf{q}_t^\top (\sum_{j \in \mathcal{I}_0} b_{ij} \mathbf{c}_j - \mathbf{c}_i)| \\
&\leq |\mathcal{T}^+| \|\sum_{j \in \mathcal{I}_0} b_{ij} \mathbf{c}_j - \mathbf{c}_i\| \\
&:= \sup_{\hat{\tau}_i} |\mathbb{E}[\hat{\tau}_i] - \mathbb{E}[\tau_i]|.
\end{aligned}
$$

Using Equation (15), we have

$$
\sup_{\hat{\tau}_i} |\mathbb{E}[\hat{\tau}_i] - \mathbb{E}[\tau_i]| \geq d_i^y |\mathcal{T}^+| / |\mathcal{T}^-|.
$$

Conservatively, we reject the estimate $\hat{\tau}_i$ if $\sup_{\hat{\tau}_i} |\mathbb{E}[\hat{\tau}_i] - \mathbb{E}[\tau_i]|$ is larger than $\delta$. That is when

$$
d_i^y > \delta |\mathcal{T}^-| / |\mathcal{T}^+|.
$$

$\square$

**Why does $d_i^y$ indicate the trustworthiness of the estimation?** Proposition 2 shows that we can control the estimation error to be below a certain threshold $\delta$ by rejecting the estimate if its error $d_i^y$ during the pre-treatment period is larger than $\delta |\mathcal{T}^-| / |\mathcal{T}^+|$. Alternatively, we can rank the estimation trustworthiness for the individuals based on $d_i^y$ alone. This is helpful when the user is willing to accept a percentage of estimations which are deemed most trustworthy. We note that this proposition only holds under the assumption that the outcomes over time are generated by the model stated in Equation 1. The outcomes generated by such a model can be nonlinear and complicated due to the representation. However, the model assumes that the outcomes over time are linear functions of the same representation. This is the reason why the pre-treatment error can be used to assess the post-treatment error. We parameterize our neural network model according to Equation 1. If it is a not good fit to the data, the model should have a large estimation error before treatment. The users should also use their domain knowledge to check if the model holds for their data, i.e., if there is any factor starting to affect the outcomes in halfway and causes the representation to change over time.

**Proposition 3** Error bound on the learned representations. Suppose that $\mathbf{v}_i = \mathbf{0}$, $\forall i \in [N]$ ($\mathbf{v}_i$ is defined in Equation 1), the total error on the learned representations for the control, i.e., the first term in the upper bound of the absolute value of the expected difference in the true and estimated ITE (R.H.S of Equation 11), is bounded as follows:

$$\sum_{j \in \mathcal{I}_0} \|\mathbf{c}_j - \tilde{\mathbf{c}}_j\| \leq \beta \mathcal{L}_s + \sum_{j \in \mathcal{I}_0} \|\xi_j\|, \tag{16}$$

where $\mathcal{L}_s$ is the supervised loss in Equation 6 and $\xi_j$ is the white noise in Equation 1.

*Proof.* We start the proof from the definition of the supervised loss.

$$
\begin{aligned}
\mathcal{L}_s &= \sum_{j \in \mathcal{I}_0} \|\tilde{\mathbf{Q}} \tilde{\mathbf{c}}_j - \mathbf{y}_j(0)\| \\
&= \sum_{j \in \mathcal{I}_0} \|\tilde{\mathbf{Q}} \tilde{\mathbf{c}}_j - (\mathbf{Q} \mathbf{c}_j + \xi_j)\| \\
&\geq \sum_{j \in \mathcal{I}_0} \left( \sum_{t \in \mathcal{T}^-} [\tilde{\mathbf{c}}_j^\top, -\mathbf{c}_j^\top] \begin{bmatrix} \tilde{\mathbf{q}}_t \\ \mathbf{q}_t \end{bmatrix} [\tilde{\mathbf{q}}_t^\top, \mathbf{q}_t^\top] \begin{bmatrix} \tilde{\mathbf{c}}_j \\ -\mathbf{c}_j \end{bmatrix} \right)^{\frac{1}{2}} - \sum_{j \in \mathcal{I}_0} \|\xi_j\|^2 \\
&\geq \tilde{\beta} \sqrt{|\mathcal{T}^-|} \sum_{j \in \mathcal{I}_0} \|\tilde{\mathbf{c}}_j - \mathbf{c}_j\| - \sum_{j \in \mathcal{I}_0} \|\xi_j\|
\end{aligned}
\tag{17}
$$

where $\tilde{\beta}$ denotes the square root of the element of the matrices $\begin{bmatrix} \tilde{\mathbf{q}}_t \\ \mathbf{q}_t \end{bmatrix} [\tilde{\mathbf{q}}_t^\top, \mathbf{q}_t^\top]$, $\forall t \in \mathcal{T}_-$, with the smallest absolute value. The first and second equations follow from Equation 6 and 1. Let $\beta$ denotes the constant $1/(\beta \sqrt{|\mathcal{T}^-|})$. Arranging the terms in inequality 17 and we prove Proposition 3. $\square$

**Justification for the supervised loss**. Proposition 3 provide a justification for the supervised loss $\mathcal{L}_s$. By optimizing the supervised loss, SyncTwin learns the representation $\tilde{\mathbf{c}}_i$ that is close to the latent variable $\mathbf{c}_i$, which also reduces the bias bound on ITE in Proposition 1.

**Rationale for the reconstruction loss**. Although the bias bounds we developed so far do not include the reconstruction loss $\mathcal{L}_c$, we believe it is useful in real applications. Our reasoning follows from the fact that unsupervised or semi-supervised loss often improve the performance of deep neural networks (Erhan et al., 2009; 2010; Hendrycks et al., 2019). In addition, the reconstruction loss ensures the representation $\tilde{\mathbf{c}}$ retains the information from the temporal covariates as required in the DAG (Figure 2). In our simulations (Section 5.1), we found that ablating the reconstruction loss leads to consistently worse performance (though the magnitude is somewhat marginal).

**Can we estimate the ITE as $\hat{\tau}_i = \mathbf{y}_i(1) - \tilde{\mathbf{Q}} \cdot \tilde{\mathbf{c}}_i$?** No, this is because $\mathcal{L}_s$ is based on the factual outcome $\mathbf{y}_i(0)$ of the control group $i \in \mathcal{I}_0$ only. For treated individuals $i \in \mathcal{I}_1$, the predictor $\tilde{\mathbf{Q}} \cdot \tilde{\mathbf{c}}_i$ can be biased for their counterfactual outcomes $\mathbf{y}_i(0)$. Hence, $\mathcal{L}_s$ is only used to learn a good representation $\tilde{\mathbf{c}}_i$ for downstream procedures, and not to directly predict counterfactual outcomes.

### A.1.2 SITUATION WITH UNOBSERVED CONFOUNDERS

In general, the unobserved confounders make it hard to provide good estimates for the ITE. The matching in pre-enrollment outcomes $\mathrm{d}_i^y$ (Equation 4) validates if the unobserved confounders $\mathbf{v}_i$ create significant error in the pre-treatment period. Using the same derivation of Theorem 1, we can see that:

$$\mathrm{d}_i^y = \|\mathbf{Q}^-(\sum_{j \in \mathcal{I}_0} b_{ij} \mathbf{c}_j - \mathbf{c}_i) + \mathbf{U}^-(\sum_{j \in \mathcal{I}_0} b_{ij} \mathbf{v}_j - \mathbf{v}_i) + \xi\|, \tag{18}$$

where $\mathbf{Q}^-$ and $\mathbf{U}^-$ are unknown but fixed matrices relating to the data generating process and $\xi$ is a term only depending on the white noise.

As shown in Proposition 1, the matching in representations encourages the first term involving $\mathbf{c}_i$ to be small. Hence, a large value in $\mathrm{d}_i^y$ implies that the remaining term involving the unmeasured confounders $\mathbf{v}_i$ is big, which leads to a large estimation error. It is worth pointing out that a small

value of $d_i^y$ does not guarantee there is no unobserved confounders — a hypothesis we cannot test empirically. For instance, consider the weights $\mathbf{U}^- = \mathbf{0}$. It follows that the second term in Equation 18 will always be zero even if $\mathbf{v}_i \neq 0$ — there exists unobserved confounders but they do not impact the outcomes before treatment (Equation 1). In summary, $d_i^y$ does not prove or disprove the *existence* of unobserved confounders; it only indicates their *impact* on the pre-treatment outcomes. Our assumption is a small relaxation of the standard no unmeasured confounders assumption by allowing a linear effect from some unmeasured confounders. More conservatively, we can assume there is no unmeasured confounders by setting all the $\mathbf{v}_i$ to $\mathbf{0}$, $\forall i \in [N]$ in Equation 1.

## A.2 Comparison of the temporal covariates allowed in the related works

As introduced in Section 2, SyncTwin is able to handle temporal covariates sampled at different frequencies, i.e. the set of observation times $\mathcal{T}_i$ and a mask $\mathbf{m}_{it}$ can be different for different individuals. In comparison, **Synthetic Control** (Abadie et al., 2010), **robust Synthetic Control** (Amjad et al., 2018), and **MC-NNM** (Athey et al., 2018) are only able to handle regularly-sampled covariates, i.e. $\mathcal{T}_i = \{-1, -2, \ldots, -L\} \; \forall i \in [N]$, and $\mathbf{m}_{it} = \mathbf{1} \; \forall i \in [N], t \in \mathcal{T}_i$. In other words, the temporal covariates $[\mathbf{x}_{is}]_{s \in [S_i]} = \mathbf{X}_i \in \mathbb{R}^{D \times S_i}$ has a matrix form.

The deep learning methods including **CRN** (Bica et al., 2020) and **RMSN** (Lim et al., 2018) have the potential to handle irregularly-measured variable-length covariates when a suitable architecture is used. However, the architectures proposed in the original papers only apply to regularly-sampled case and no simulation or real data experiments were conducted for the more general irregular cases.

## A.3 Comparison of the causal assumptions in the related works

Table 3: Comparison of the causal assumptions in the related works. The definitions of Consistency, Sequential overlap, and No unobserved confounder are given in A.3 in bold. The data generating model (D.G.M) in Equation 1 contains the one in Equation 9 as a special case.

| Approach | Ref | Consistency | D.G.M | Sequential Overlap | No unobserved conf. |
|---|---|---|---|---|---|
| SC | Abadie (2019) | Yes | Equation 19 | - | - |
| RSC | Amjad et al. (2018) | Yes | Equation 19 | - | - |
| MC-NNM | Athey et al. (2018) | Yes | Equation 19 | - | - |
| CRN | Bica et al. (2020) | Yes | - | Yes | Yes |
| RMSN | Lim et al. (2018) | Yes | - | Yes | Yes |
| SyncTwin | This work | Yes | Equation 1 | - | - |

### A.3.1 Synthetic control

As shown in Table 3, Synthetic control (Abadie et al., 2010; Abadie, 2019) and its variants (Athey et al., 2018; Amjad et al., 2018) rely on two causal assumptions: (1) **consistency**: $\mathbf{y}_{it}(\mathbf{a}_{it}) = \mathbf{y}_{it}$ and (2) **data generating assumption** (linear factor model):

$$\mathbf{y}_{it}(0) = \mathbf{q}_t^\top \underline{\mathbf{x}}_i + \mathbf{u}_t^\top \mathbf{v}_i + \xi_{it} \quad \forall i \in [N], \; t \in \mathcal{T}^- \cup \mathcal{T}^+. \tag{19}$$

where $\underline{\mathbf{x}}_i = \text{vec}(\mathbf{X}_i) \in \mathbb{R}^{D \times L}$, vec is the vectorization operation; $\mathbf{u}_t \in \mathbb{R}^U$ and $\mathbf{q}_t \in \mathbb{R}^{D \times L}$ are time-varying variables and $\mathbf{v}_i \in \mathbb{R}^U$ is a latent variable. $\xi_{it}$ is an error term that has mean zero and satisfies $\xi_{it} \perp\!\!\!\perp \mathbf{a}_{rs}, \mathbf{x}_r, \mathbf{u}_s, \mathbf{v}_r$ for $\forall \, k, r, s, t$.

It is worth highlighting that the data generating assumption of Synthetic Control is a **special case** of the more general assumption of SyncTwin in Equation 1. To see this, let $\mathbf{c}_i = \underline{\mathbf{x}}_i = \text{vec}(\mathbf{X}_i)$ in Equation 1, i.e. we use the flattened temporal covariates directly as the representation. Further let $\phi_\theta(\mathbf{c}_i, t_{is}) = \mathbf{c}_i[Ds : D(s+1)]$ and $\varepsilon_{is} = 0$, where $\mathbf{c}[a:b]$ takes a slice of vector $\mathbf{c}$ between index $a$ and $b$. The result is exactly Equation 19.

**Why does Synthetic Control tend to over-match**? Both SyncTwin and Synthetic Control estimate the treatment effects using a weighted combination of control outcomes (Equation 3). However, Synthetic Control finds weight $b_{ij}$ in a different way by directly minimizing

$$\mathcal{L}_x = ||\underline{\mathbf{x}}_i - \sum_j b_{ij} \underline{\mathbf{x}}_j||.$$

Since $\underline{\mathbf{x}}_i$ contains the observation noise and other random components that do not relate to the outcomes, the weights $b_{ij}$ that minimize $\mathcal{L}_x$ tend to over-match, i.e. they capture the irrelevant randomness in $\underline{\mathbf{x}}_i$. In contrast, SyncTwin finds $b_{ij}$ based on the learned representations $\tilde{\mathbf{c}}_i$ rather than $\underline{\mathbf{x}}_i$ ($\mathcal{L}_m$, Equation 6). Since $\tilde{\mathbf{c}}_i$ has much lower dimensionality than $\underline{\mathbf{x}}_i$, the reconstruction loss $\mathcal{L}_r$ encourages the Seq2Seq network to learn a $\tilde{\mathbf{c}}_i$ that only retains the signal in $\underline{\mathbf{x}}_i$ but not the noise. Meanwhile, the supervised loss encourages $\tilde{\mathbf{c}}_i$ to only retain the information that predicts the outcomes. As a consequence, we expect the weights based on $\tilde{\mathbf{c}}_i$ to be less prone to over-match. Moreover, since the relationship between $\tilde{\mathbf{c}}_i$ and $\underline{\mathbf{x}}_i$ is nonlinear (as captured by the decoder network), the weights $b_{ij}$ that minimize $\mathcal{L}_m$ will generally not minimize the Synthetic Control objective $\mathcal{L}_x$, therefore avoiding over-match.

### A.3.2 COUNTERFACTUAL RECURRENT NEURAL NETWORKS

As shown in Table 3, CRN (Bica et al., 2020) and RMSN (Lim et al., 2018) makes the following three causal assumptions. (1) **Consistency**: $\mathbf{y}_{it}(\mathbf{a}_{it}) = \mathbf{y}_{it}$. (2) **Sequential overlap** (aka. positivity): $Pr(\mathbf{a}_{it} = 1|\mathbf{a}_{i,t-1}, \mathbf{x}_{it}) > 0$ whenever $Pr(\mathbf{a}_{i,t-1}, \mathbf{x}_{it}) \neq 0$. (3) **No unobserved confounders**: $\mathbf{y}_{it}(0), \mathbf{y}_{it}(1) \perp\!\!\!\perp \mathbf{a}_{it} \mid \mathbf{x}_{it}, \mathbf{a}_{i,t-1}$. In summary, CRN makes the same consistency assumption as SyncTwin. However, SyncTwin does not assume sequential overlap or no unobserved confounders while CRN does not make assumptions on the data generating model.

The sequential overlap assumption means that the individuals should have non-zero probability to change treatment status at any time $t \geq 0$ given the history. This assumption is violated in the clinical observational study setting we outlined in Section 1, where the treatment group will continue to be treated and cannot switch to the control group after they are enrolled (and similarly for the control group). While the sequential overlap assumption allows these methods to handle more general situations where treatment switching do occur, their performance is negatively impacted in the "special" (yet still widely applicable) setting we consider in this work.

While CRN makes strict no-unobserved-confounder assumption, SyncTwin allows certain types of unobserved confounders to occur. In particular, the latent factor $\mathbf{v}_i$ in Equation 1 can be unobserved confounders. Being less reliant on no-unobserved-confounder assumption is important for medical applications because it is hard to assume the dataset captures all aspects of the patient health status. SyncTwin 's ability to handle unobserved confounders $\mathbf{v}_i$ relies on the validity of its data generating assumption, which we discuss next.

**Why does SyncTwin not explicitly require overlap**? The overlap assumption is commonly made in treatment effect estimation methods. We first give a very brief review of why two importance classes of methods need overlap. (1) For methods that rely on propensity scores, overlap makes sure that the propensity scores are not zero, which enables various forms of propensity weighting. (2) For methods that rely on covariate adjustment, overlap ensures that the conditional expectation $\mathbb{E}[\mathbf{y}_i|\mathbf{X}_i, \mathbf{a}_i]$ is well-defined, i.e. the conditioning variables $(\mathbf{X}_i, \mathbf{a}_i)$ have non-zero probability. In comparison, SyncTwin relies on neither the propensity scores nor the explicit adjustment of covariates, and hence it does not make overlap assumption *explicitly*. However, as discussed in Proposition 1, SyncTwin requires the synthetic twin to match the representations $\mathbf{d}_i^c \approx 0$, which implies $\tilde{\mathbf{c}}_i \approx \sum_{j \in \mathcal{I}_0} b_{ij}\tilde{\mathbf{c}}_i$ for some $b_{ij}$ — the target individual should be in or close to the convex hull formed by the controls in the representation space. This condition has a similar spirit to overlap (but very different mathematically). When overlap is satisfied there tend to be control individuals in the neighbourhood of the treated individual, making it easier to construct matching twins. Conversely, if overlap is violated, the controls will tend to far away from the treated individual, making it harder to construct a good twin.

### A.4 THE GENERALITY OF THE ASSUMED DATA GENERATING MODEL

SyncTwin assumes that the outcomes are generated by a *latent* factor model (Teräsvirta et al., 2010) with the *latent* factors $\mathbf{c}_i$ learnable from covariates $\mathbf{X}_i$ and the *latent* factors $\mathbf{v}_i$ that are unobserved confounders. We assume the dimensionality of $\mathbf{c}_i$ and $\mathbf{v}_i$ to be low compared with the number of time steps. Despite its seemingly simple form, the assumed latent factor model is very flexible because the factors are in fact *latent variables*.

The latent factor model is widely studied in Econometrics. In many real applications, the temporally observed variables naturally have a low-rank structure, thus can be described as a latent factor model (Abadie & Gardeazabal, 2003; Abadie et al., 2010). The latent factor model also captures many

of well-studied scenarios as special cases (Finkel, 1995) such as the conventional additive unit and time fixed effects ($y_{it}(0) = q_t + c_i$). Last but not least, It has also been shown that the low-rank latent factor models can well approximate many nonlinear latent variable models (Udell & Townsend, 2017).

Latent factor models in the static setting are very familiar in the deep learning literature. Consider a deep feed-forward neural network that uses a linear output layer to predict some real-valued outcomes $y \in \mathbb{R}^D$ in the static setting (notations used in this example are not related to the ones used in the rest of the paper). Denote the last layer of the neural network as $h_{-1} \in \mathbb{R}^K$; it is easy to see that the neural network corresponds to a latent factor model i.e. $y = Ah_{-1} + b$, where $h_{-1}$ is the latent factor. Note that this holds true for arbitrarily complicated feed-forward networks as long as the output layer is linear.

### A.5 Estimating ITE for control and new individuals

We have been focusing on estimating ITE for a treated individual $i \in \mathcal{I}_1$. The same approach can estimate the ITE for a control individual without loss of generality. After obtaining the representation $\tilde{c}_i$ for $i \in \mathcal{I}_0$, SyncTwin can use the treatment group $j \in \mathcal{I}_1$ to construct the synthetic twin by optimizing the matching loss Equation 8. The checking and estimation procedure remains the same.

SyncTwin can also estimate the effect of a new individual $i \notin [N]$. The same idea still applies, but this time we need to construct two synthetic twins: one from the control group and one from the treatment group. The ITE estimation can be obtain using the difference between the two twins.

SyncTwin also easily generalizes to the situation where there are $A > 1$ treatment groups each receiving a different treatment. In this case, the treatment indicator $a_i \in [0, 1, \ldots, A]$. For a target individual in *any* of the treatment groups, SyncTwin can construct its twin using the control group $\mathcal{I}_0$. The remaining steps are the same as the single treatment group case.

### A.6 Unrelated works with similar terminology

Several recent works in the deep learning ITE literature employ similar terminologies such as "matching" (Johansson et al., 2018; Kallus, 2018). However, they are fundamentally different from SyncTwin because they only work for static covariates and they try to match the overall distribution of the treated and control group rather than constructing a synthetic twin that matches one particular treated individual.

The Virtual Twin method (Foster et al., 2011) is designed for randomized clinical trials where there is no confounding (temporal or static). As a result, it cannot overcome the confounding bias when the problem is to estimate causal treatment effect from *observational data*.

### A.7 Implementation details of the benchmark algorithms

**Synthetic control**. We used the implementation of Synthetic Control in the `R` package `Synth` (1.1-5). The package is available at `https://CRAN.R-project.org/package=Synth`.

**Robust Synthetic Control**. We used the implementation accompanied with the original paper (Amjad et al., 2018) at `https://github.com/SucreRouge/synth_control`. We optimized the hyperparameters on the validation set using the method described in Section 3.4.3 Amjad et al. (2018). The best hyperparameter setting was then applied to the test set.

**MC-NNM**. We used the implementation in the `R` package `SoftImpute` (1.4) available at `https://CRAN.R-project.org/package=softImpute`. The regularization strength $\lambda$ is tuned on validation set using grid search before applied to the testing data.

**Counterfactual Recurrent Network** and **Recurrent Marginal Structural Network**. We used the implementations by the authors Bica et al. (2020); Lim et al. (2018) at `https://bitbucket.org/mvdschaar/mlforhealthlabpub/src/master/`. The networks were trained on the training dataset. We experimented different hyper-parameter settings on the validation dataset, and applied the best setting to the testing data. We also found that the results are not sensitive to the hyperparameters.

**Counterfactual Gaussian Process**. We used the implementation with GPy (GPy, since 2012), which is able to automatically optimize the hyperparameters such as the kernel width using the validation data.

**One-nearest neighbour**. We used our own implementation. Since no parameters need to be learned or tuned, the algorithm was directly applied on the testing dataset.

Search range of hyper-parameters

1. Synthetic control: hyperparameters are optimized by `Synth` directly.
2. Robust Synthetic control: num_sc $\in \{1, 2, 3, 4, 5\}$
3. MC-NNM: $C \in \{3, 4, 5, 8, 10\}$
4. Counterfactual Recurrent Network: max_alpha $\in \{0.1, 0.5, 0.8, 1\}$, hidden dimension $H \in \{32, 64, 128\}$
5. Recurrent Marginal Structural Network: hidden dimension $H \in \{32, 64, 128\}$
6. Counterfactual Gaussian Process: hyperparameters are optimized by `GPy` directly.

## A.8 DETAILED TRAINING, VALIDATION AND INFERENCE PROCEDURE

As is standard in machine learning, we perform model training, validation and inference (testing) on three disjoint datasets, $\mathcal{D}^{tr}$, $\mathcal{D}^{va}$ and $\mathcal{D}^{te}$. We use $\mathcal{D}_0^{tr}$ and $\mathcal{D}_1^{tr}$ to denote the control and the treated in the training data and use similar notations for validation and testing data.

**Training**. On the training dataset $\mathcal{D}_0^{tr}$, we learn the representation networks by optimizing $\mathcal{L}^{tr} = \lambda_r \mathcal{L}_r + \lambda_p \mathcal{L}_s$, where $\mathcal{L}_r$ and $\mathcal{L}_s$ are the loss functions defined in Equation 6. The hyperparameter $\lambda_r$ and $\lambda_p$ controls the relative importance between the two losses. We provide an ablation study in Section 5.1 and perform detailed analysis on hyperparameter importance in Appendix A.13. The objective $\mathcal{L}^{tr}$ can be optimized using stochastic gradient descent. In particular, we used the ADAM algorithm with learning rate 0.001 (Kingma & Ba, 2014).

**Validation**. Since we never observe the true ITE, we cannot evaluate the error of ITE estimation, $||\tau_i - \hat{\tau}_i||_2^2$. As a standard practice (Bica et al., 2020), we rely on the *factual* loss on observed outcomes: $\mathcal{L}^{va} = \sum_{j \in \mathcal{D}_0^{va}} ||\mathbf{y}_i(0) - \hat{\mathbf{y}}_i(0)||_2^2$, where $\hat{\mathbf{y}}_i(0)$ is defined as in Equation 2 and obtained as follows. We obtain the $\tilde{\mathbf{c}}_i$ for all $i \in \mathcal{D}^{va}$ and then optimize the matching loss $\mathcal{L}_m(\mathcal{D}_0^{va}, \mathcal{D}_1^{va})$ to find weights $\mathbf{b}_i^{va}$. It is important to keep the encoder fixed throughout the optimization; otherwise it might overfit to $\mathcal{D}^{va}$. Finally, $\hat{\mathbf{y}}_i(0) = \sum_{j \in \mathcal{D}_0^{va}} b_{ij}^{va} \mathbf{y}_j(0)$.

**Inference**. The first steps of the inference procedure are the same as validation. We start by obtaining the representation $\tilde{\mathbf{c}}_i$ for all $i \in \mathcal{D}^{te}$ and then obtain weights $\mathbf{b}_i^{te}$ by optimizing the matching loss $\mathcal{L}_m(\mathcal{D}_0^{te}, \mathcal{D}_1^{te})$ while keeping the encoder fixed. Using weights $\mathbf{b}_i^{te}$, the ITE for any $i \in \mathcal{D}_1^{te}$ can be estimated as $\hat{\tau}_i = \mathbf{y}_i(1) - \sum_{j \in \mathcal{D}_0^{te}} b_{ij}^{te} \mathbf{y}_j(0)$ according to Equation 3. Similarly, we obtain $\hat{\mathbf{c}}_i$, $\hat{\mathbf{y}}_{it}(0)$ according to in Equation 2. The expert can check $\mathbf{d}_i^y$ to evaluate the trustworthiness of $\hat{\tau}_i$.

Table 4: Parameters for each component of the architecture and the loss function for training each parameter.

| Component | Parameters | Loss function | Reference |
|---|---|---|---|
| Attentive Encoder | GRU-D parameters, $\mathbf{b}$ | $\mathcal{L}_s, \mathcal{L}_r$ | Section 4.1 |
| Decoder | LSTM parameters, $\mathbf{k}_0, \mathbf{w}_0, \mathbf{k}_1, \mathbf{W}_1$ | $\mathcal{L}_r$ | Section 4.1 |
| Linear outcome prediction | $\tilde{\mathbf{Q}}$ | $\mathcal{L}_s$ | Section 4.1 |
| Weights | $\mathbf{B}$ | $\mathcal{L}_m$ | Section 4.2 |

## A.9 OPTIMIZING THE MATCHING LOSS

Here we present a way to optimize the matching loss $\mathcal{L}_m$ in Equation 8. To ensure the three constraints discussed in Section 4.2 while also allowing gradient-based learning algorithm, we reparameterize $\mathbf{b}_i = \text{Gumbel-Softmax}(f_m(\mathbf{z}_i), \tau)$, where $\mathbf{z}_i \in \mathbb{R}^{N_0}$, $f_m(\cdot)$ is a masking function that sets the element $z_{ii} = -\text{Inf}$ to satisfy constraint (3). Gumbel-Softmax$(\cdot, \tau)$ is the Gumbel softmax function

**Algorithm 1:** SyncTwin training procedure.

---

**Input:** Training data set: $\mathcal{D}_0^{tr}, \mathcal{D}_1^{tr}$
**Input:** Hyperparameters: $\lambda_r, \lambda_p$
**Input:** Encoder, Decoder, $\tilde{\mathbf{Q}}$
**Input:** Training iteration $max\_itr$, batch size $batch\_size$, Optimizer
**Output:** Trained Encoder, Decoder and $\tilde{\mathbf{Q}}$
Randomly initialize Encoder and Decoder; set $\tilde{\mathbf{Q}} = \mathbf{0}$
**for** $itr \in (0, max\_itr]$ **do**

> Randomly draw a mini-batch of control units $\mathcal{D}_0 \subset \mathcal{D}_0^{tr}$ with $batch\_size$ samples.
> Randomly draw a mini-batch of treated units $\mathcal{D}_1 \subset \mathcal{D}_1^{tr}$ with $batch\_size$ samples.
> Evaluate training loss $\mathcal{L}^{tr}(\mathcal{D}_0, \mathcal{D}_1) = \lambda_r \mathcal{L}_r(\mathcal{D}_0, \mathcal{D}_1) + \lambda_p \mathcal{L}_s(\mathcal{D}_0)$ (defined in Equation 6)
> Calculate the gradient of $\mathcal{L}^{tr}(\mathcal{D}_0, \mathcal{D}_1)$ via back propagation.
>
> Update all encoder, decoder parameters and $\tilde{\mathbf{Q}}$ using the Optimizer

---

**Algorithm 2:** SyncTwin inference procedure.

---

**Input:** Testing data set: $\mathcal{D}_0^{te}, \mathcal{D}_1^{te}$
**Input:** Trained Encoder
**Input:** Training iteration $max\_itr$, batch size $batch\_size$, Optimizer
**Output:** Estimated ITE $\hat{\tau}_i, \forall i \in \mathcal{D}_1^{te}$
Initialize a size $|\mathcal{D}_1^{te}|$ by $|\mathcal{D}_0^{te}|$ matrix $\mathbf{B} = \mathbf{0}$ as the weight matrix.
Use Encoder to get representation $\tilde{\mathbf{c}}_i, \forall i \in \mathcal{D}_0^{te} \cup \mathcal{D}_1^{te}$
**for** $itr \in (0, max\_itr]$ **do**

> Randomly draw a mini-batch of treated units $\mathcal{D}_1 \subset \mathcal{D}_1^{tr}$ with $batch\_size$ samples.
> Evaluate matching loss $\mathcal{L}_m(\mathcal{D}_0^{te}, \mathcal{D}_1)$ (defined in Equation 8)
> Calculate the gradient of $\mathcal{L}_m(\mathcal{D}_0^{te}, \mathcal{D}_1)$ via back propagation.
> Update $\mathbf{B}$ using the Optimizer while keeping the Encoder fixed.

Use weight matrix $\mathbf{B}$ to obtain $\hat{\tau}_i, \forall i \in \mathcal{D}_1^{te}$ using Equation 3.

---

with temperature hyper-parameter $\tau$ (Jang et al., 2016). It is straightforward to verify that $\mathbf{b}_k$ satisfies the three constraints while the loss $\mathcal{L}_m$ remains differentiable with respect to $\mathbf{z}_k$. We use the Gumbel softmax function instead of the standard softmax function because Gumbel softmax tend to produce sparse vector $\mathbf{b}_k$, which is highly desirable as we discussed in Section 4.

The memory footprint to directly optimize $\mathcal{L}_m$ is $O\big((|\mathcal{D}_0| + |\mathcal{D}_1|) \times |\mathcal{D}_0|\big)$, which can be further reduced to $O\big(|\mathcal{D}_B| \times |\mathcal{D}_0|\big)$ if we use stochastic gradient decent with a mini-batch $\mathcal{D}_B \subseteq \mathcal{D}_0 \cup \mathcal{D}_1$.

## A.10 THE SIMULATION MODEL

In Equation 9, $R_t$ is the LDL cholesterol level (outcome) and $I_t$ is the dosage of statins. For each individual in the treatment group, one dose of statins (10 mg) is administered daily after the treatment starts, which gives dosage $I_t = 0$ if $t \leq t_0$ and $I_t = 1$ otherwise. $K$, $H$ and $D_{50}$ are constants fixed to the values reported in Faltaos et al. (2006). $K_t^{in} \in \mathbb{R}$ is a individual-specific time varying variable that summarizes a individual's physiological status including serum creatinine, uric acid, serum creatine phosphokinase (CPK), and glycaemia. $P_t$ and $D_t$ are two intermediate temporal variables both affecting $R_t$.

## A.11 ADDITIONAL SIMULATION RESULTS

Table 5: Mean absolute error on ITE with varying irregular $m$. $S = 25$ and $p_0 = 0.5$ are used in all cases. Estimated standard deviations are shown in the parentheses. The best performer is in bold. * did not finish within 48h.

| Method | $N_0 = 200$ | | | $N_0 = 1000$ | | |
|---|---|---|---|---|---|---|
| | $m = 0.7$ | $m = 0.5$ | $m = 0.3$ | $m = 0.7$ | $m = 0.5$ | $m = 0.3$ |
| SyncTwin-Full | **0.129** (.008) | **0.142** (.009) | **0.190** (.012) | **0.109** (.006) | **0.116** (.006) | **0.141** (.008) |
| SyncTwin-$\mathcal{L}_r$ | 0.158 (.012) | 0.176 (.014) | 0.245 (.017) | 0.125 (.007) | 0.133 (.008) | 0.175 (.011) |
| SyncTwin-$\mathcal{L}_s$ | 0.129 (.009) | 0.152 (.010) | 0.234 (.016) | 0.139 (.009) | 0.134 (.009) | 0.172 (.012) |
| SC | 0.155 (.017) | 0.201 (.015) | 0.326 (.023) | 0.145 (.015) | 0.215 (.020) | 0.359 (.026) |
| RSC | 0.414 (.021) | 0.520 (.028) | 0.639 (.043) | * | 0.495 (.028) | * |
| MC-NNM | 0.363 (.020) | 0.556 (.031) | 0.898 (.050) | 0.174 (.010) | 0.332 (.021) | 0.556 (.036) |
| CFRNet | 0.317 (.030) | 0.303 (.018) | 0.481 (.030) | 0.143 (.009) | 0.187 (.012) | 0.255 (.018) |
| CRN | 0.300 (.020) | 0.364 (.026) | 0.424 (.028) | 0.416 (.027) | 0.456 (.029) | 0.677 (.040) |
| RMSN | 0.327 (.025) | 0.338 (.025) | 0.391 (.026) | 0.381 (.031) | 0.400 (.030) | 0.471 (.031) |
| CGP | 0.568 (.037) | 0.553 (.037) | 0.631 (.045) | 0.605 (.039) | 0.626 (.039) | 0.689 (.044) |
| 1NN | 1.584 (.080) | 1.725 (.098) | 1.703 (.096) | 1.455 (.083) | 1.680 (.088) | 1.531 (.089) |

Table 6: Mean absolute error on ITE under different lengths of the temporal covariates $S$. $m = 1$ and $p_0 = 0.5$ are used in all cases. Estimated standard deviations are shown in the parentheses. The best performer is in bold. * did not finish within 48 hours.

| Method | $N_0 = 200$ | | | $N_0 = 1000$ | | |
|---|---|---|---|---|---|---|
| | $S = 15$ | $S = 25$ | $S = 45$ | $S = 15$ | $S = 25$ | $S = 45$ |
| SyncTwin-Full | **0.123** (.008) | **0.119** (.008) | **0.112** (.007) | **0.097** (.005) | **0.093** (.005) | **0.085** (.004) |
| SyncTwin-$\mathcal{L}_r$ | 0.136 (.010) | 0.139 (.010) | 0.139 (.010) | 0.114 (.006) | 0.101 (.006) | 0.098 (.006) |
| SC | 0.139 (.018) | 0.149 (.018) | 0.138 (.021) | 0.190 (.029) | 0.214 (.036) | 0.215 (.044) |
| RSC | 0.348 (.023) | 0.321 (.018) | 0.228 (.011) | * | 0.302 (.014) | * |
| MC-NNM | 0.454 (.023) | 0.226 (.011) | 0.159 (.008) | 0.139 (.007) | 0.124 (.006) | 0.109 (.005) |
| CFRNet | 0.126 (.006) | 0.130 (.007) | 0.143 (.008) | 0.113 (.006) | 0.106 (.007) | 0.095 (.005) |
| CRN | 0.283 (.019) | 0.360 (.024) | 0.422 (.022) | 0.387 (.024) | 0.357 (.021) | 0.426 (.023) |
| RMSN | 0.331 (.024) | 0.332 (.026) | 0.399 (.024) | 0.363 (.029) | 0.385 (.032) | 0.447 (.029) |
| CGP | 0.561 (.036) | 0.561 (.035) | 0.549 (.035) | 0.578 (.037) | 0.602 (.038) | 0.611 (.038) |
| 1NN | 1.356 (.072) | 1.614 (.078) | 1.575 (.078) | 1.322 (.072) | 1.384 (.083) | 1.744 (.098) |

Table 5 shows the results under irregularly-measured covariates with varying degree of irregularity $m$ (smaller $m$, more irregular and fewer covariates are observed). For methods that are unable to

deal with irregular covariates, we first impute the unobserved values using Probabilistic PCA before applying the algorithms (Hegde et al., 2019). SyncTwin achieves the best performance in all cases. Furthermore, SyncTwin's performance deteriorates more slowly than the benchmarks when sampling becomes more irregular (larger $m$). This suggests that the encoder network in SyncTwin is able to learn good representations even from highly irregularly-measured sequences. Table 6 shows the results under various lengths of the observed covariates $S$ (smaller $S$, shorter sequences are observed). Again SyncTwin achieves the best performance in all cases. As expected, SyncTwin makes smaller error when the observed sequence is longer. Note that this is not the case of CRN and RMSN — their performance deteriorates when the observed sequence is longer. This might indicate that these two methods are less able to learn good balancing representations (or balancing weights) when the sequence is longer.

## A.12 Sparsity compared with Synthetic Control

In Figure 4 we have shown visualy that SyncTwin produces sparser solution than SC. To quantify the differences, we report the Gini index ($\sum_{ij} b_{ij}(1 - b_{ij})/N_1$), entropy ($\sum_{ij} -b_{ij}\log(b_{ij})/N_1$) and the number of contributors used to construct the twin ($\sum_{ij} \mathbf{1}\{b_{ij} > 0\}/N_1$) in the simulation study. All three metrics reflect the sparsity of the learned weight vector (smaller more sparse). Table 7 shows that SyncTwin achieve sparser results that SC in all metrics considered. The full and ablated versions of SyncTwin have similar sparsity because the sparsity is regulated in the matching loss, which all versions share. It is worth pointing out that RSC and MC-NNM do not produce sparse weights and the weights do not need to be positive and sum to one (Amjad et al., 2018; Athey et al., 2018).

Table 7: Sparsity metrics of the learned $\mathbf{b}_i$. Estimated standard deviations are shown in the parentheses. Here $p_0 = 0.5$, $m = 1$, $S = 25$. The worst performer is italicized

| Method | $N_0 = 200$ | | | $N_0 = 1000$ | | |
|---|---|---|---|---|---|---|
| | Gini | Entropy | N Control | Gini | Entropy | N Matched |
| SyncTwin-Full | 0.213 (.016) | 0.409 (.030) | 1.755 (.069) | 0.242 (.017) | 0.483 (.034) | 1.830 (.073) |
| SyncTwin-$\mathcal{L}_r$ | 0.214 (.017) | 0.407 (.033) | 1.780 (.075) | 0.267 (.018) | 0.548 (.037) | 1.930 (.080) |
| SyncTwin-$\mathcal{L}_s$ | 0.213 (.016) | 0.409 (.030) | 1.760 (.068) | 0.249 (.018) | 0.500 (.037) | 1.930 (.083) |
| SC | *0.792 (.009)* | *1.871 (.035)* | *6.125 (.135)* | *0.862 (.006)* | *2.274 (.029)* | *7.059 (.110)* |
| RSC | - | - | 1.903 (.084) | - | - | 2.311 (.964) |

## A.13 Sensitivity of Hyper-Parameters

It is beneficial to understand the network's sensitivity to each hyper-parameter so as to effectively optimize them during validation. In addition to the standard hyper-parameters in deep learning (e.g. learning rate, batch size, etc.), SyncTwin also includes the following specific hyper-parameters: (1) $\tau$, the temperature of the Gumbel-softmax function Appendix A.9, (2) $\lambda_p$ in the training loss $\mathcal{L}^{tr}$ (since only the ratio between $\lambda_p$ and $\lambda_r$ matters, we keep $\lambda_r = 1$ and search different values of $\lambda_p$) , and (3) $H$, the dimension of the representation $\tilde{\mathbf{c}}_i$.

Here we present a sensitivity analysis on the hyper-parameters $H$, $\lambda_p$ and $\tau$ using the simulation framework detailed in Section 5.1. Here we present the results for $N_0 = 2000$ and $S = 15$ although these results generalize to all the simulation settings we considered. The results are presented in Figure 6, where we can derive two insights.

Firstly, the hyper-parameter $\tau$ is very important to the performance and need to be tuned carefully during validation. This is understandable because $\tau$ is the temperature parameter of the Gumbel softmax function and it directly controls the sparsity of matrix $\mathbf{B}$. In comparison, hyper-parameter $H$ and $\lambda_p$ do not impact the performance in significant way. Therefore we recommend to use $H = 40$ and $\lambda_p = 1$ as the default.

Secondly, we observe that the validation loss $\mathcal{L}^{va}$ closely tracks the error on ITE estimation (which is not directly observable in reality). These results support the use of $\mathcal{L}^{va}$ to validate models and perform hyper-parameter optimization.

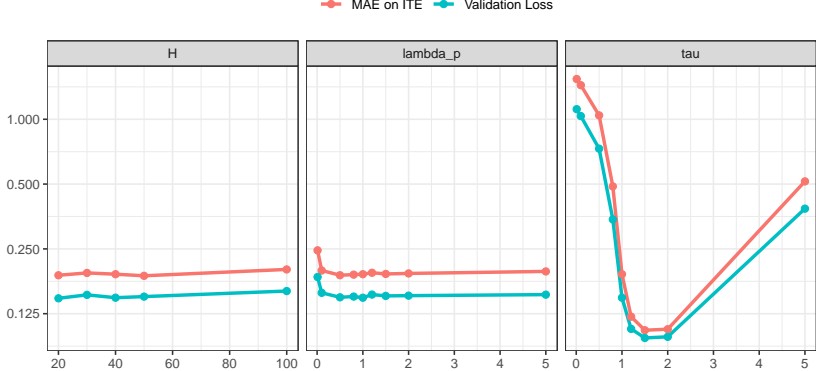

Figure 6: The sensitivity of hyper-parameters on the mean absolute error of ITE estimation and the validation loss defined in Section 4.3. The left panel shows the results for various choices of $H$; the middle panel shows $\lambda_p$; and the right panel shows $\tau$. The y-axis is shown in log scale.

## A.14 COMPUTATION TIME

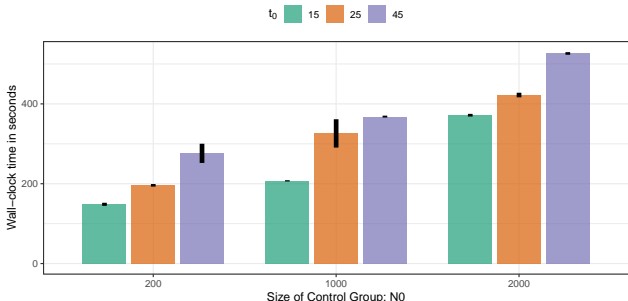

Figure 7: The wall-clock time of the simulation study under different settings. For each setting, 10 independent simulation runs were conducted. The bar shows the average wall-clock time and the line range captures the 95% confidence interval.

In figure 7 we present the wall-clock computation time (in seconds) of SyncTwin under various simulation conditions — with the control group size $N_0 = (200, 1000, 2000)$ and the length of pre-enrollment period $S = (15, 25, 45)$. The simulations were performed on a server with a Intel(R) Core(TM) i5-8600K CPU @ 3.60GHz and a Nvidia(R) GeForce(TM) RTX 2080 Ti GPU. All simulations finished within 10 mins. As we expect, the computation time increases with respect to $N_0$ and $S$ as more data need to be processed. However, a 10-fold increase in $N_0$ only approximately doubled the computation time, suggesting that SyncTwin scales well with sample size. In comparison, $S$ seems to affect the computation time more because the encoder and decoder need to be trained on longer sequences.

## A.15 ADDITIONAL RESULTS IN THE CPRD STUDY

We the treatment and the control group in the CPRD experiment are selected based on the selection criterion in Figure 8. We have followed all the guidelines listed in Dickerman et al. (2019) to make sure the selection process does not increase the confounding bias. The summary statistics of the treatment and control groups are listed below. We can clearly see a selection bias as the treatment group contains a much higher proportion of male and people with previous cardiovascular or renal diseases.

Table 8: The summary statistics of the treatment and control groups

|  | Treatment Group | Control Group |
|---|---|---|
| % male | 59% | 51% |
| Median age | 61 | 60 |
| Median Townsend Index | 8 | 8 |
| % CVD | 16% | 9% |
| % Renal disease | 16% | 12% |
| % Atrial Fibrillation | 4% | 4% |

## A.16 COHORT SELECTION CRITERION IN THE CPRD STUDY

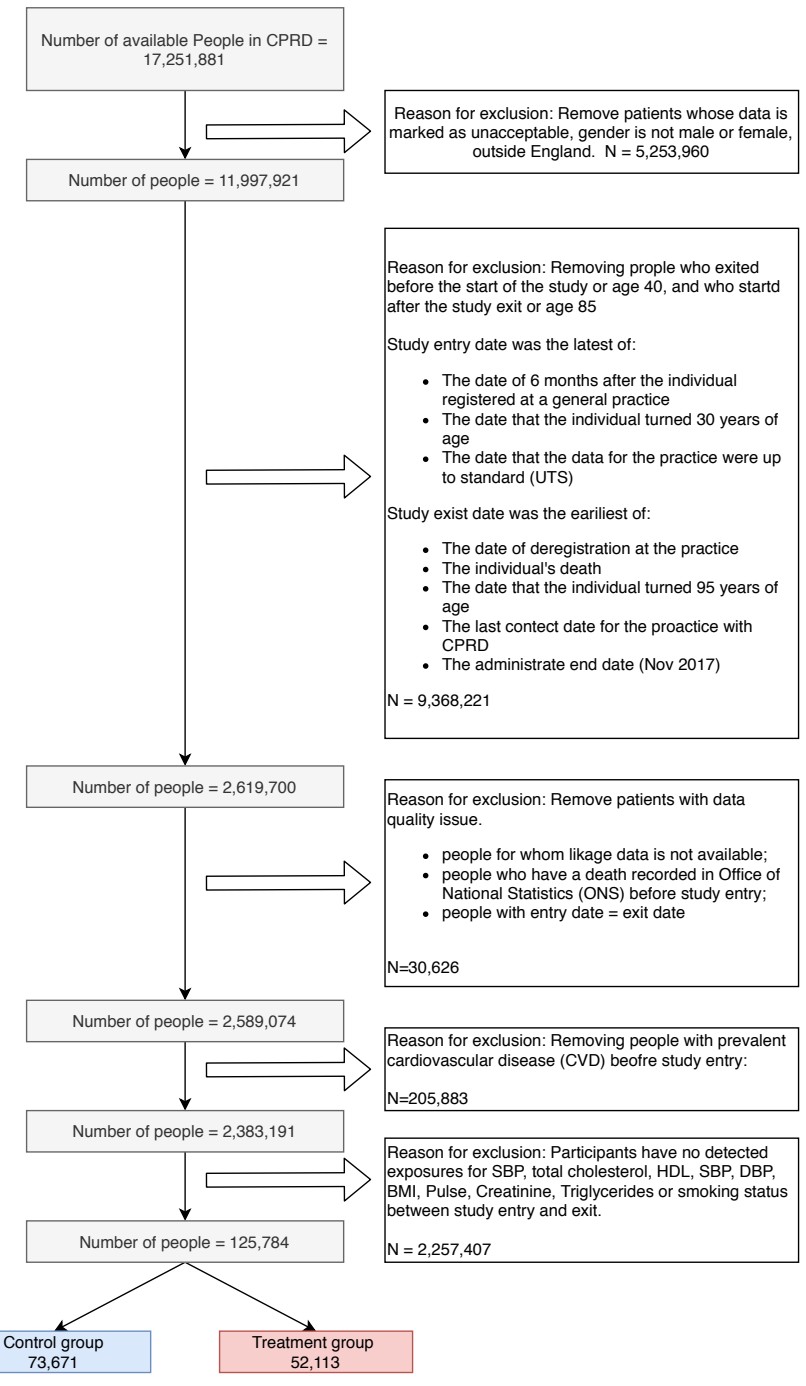

Figure 8: Flowchart for selection of eligible individuals from CPRD for the observational study on the treatment effect of statins. Numbers represent unique individuals in each group.

