# OpenReview forum: "SyncTwin: Transparent Treatment Effect Estimation under Temporal Confounding"
_ICLR.cc/2021/Conference — Reject_

### Official Review · AnonReviewer2 · 2020-10-28
**interesting work on synthetic control on representations; requires some expanded discussion**

**Rating:** 7
**Confidence:** 4

**Review:**

I enjoyed reading this paper and think it tackles an important problem. The paper works with a particular model of the data generation and propose to do synthetic control in representation space which ensures that the considered data generation/model are not inflexible.

I list a few issues here; they are mainly about the discussion of the assumptions.

There is very little discussion about identification guarantees in the paper. While A.1 provides some justification in this regard, I would've liked to see a paragraph about the identification guarantees for the proposed algorithm.

It seems to me that the functional linearity assumption of the model is what allows the method to not require sequential overlap. Could the authors expand on this?

I believe that the key requirement of the method is the linearity of the control outcome in the latent c_i even when it is a non-linear function of the covariates; I think this helps the method not face serious non-identification issues. Could the authors clarify this intuition?

I liked the idea of using the control outcome as a way to understand whether there is unobserved confounding as per the modelling assumption. The authors should however clearly state that no-unobserved-confounding is an assumption they also require to estimate effects; effect estimation is in generally impossible if v_i exists and is unobserved.

I liked the experimental section also. The consistently better performance (even over the ablated models) provided some good evidence for the merit of the method. The real experiment (and the transperancy discussion) provided further strong evidence in favor of synctwin.

Finally, it would've been nice to see one more synthetic experiment but seeing that the data generation considered in the paper is a biologically informed model (not a toy one), I'm happy with them for now.

I'm happy to increase my score if the discussion around assumptions is sufficiently clarified.

---

> ### Author Response · Authors · 2020-11-17
> **Reply to Reviewer 2**
>
> Thank you for the excellent comments and suggestions; we have updated the paper after taking all your comments into account. Please refer to the point-by-point response below and let us know if anything needs further clarification.
>
> **Identification guarantee**
>
> To better clarify the issue around identification, we have significantly expanded the discussions in Section 4 and Appendix A.1 as well as adding a new Proposition 2.
>
> The identification guarantee is closely related to two key variables: $d^c$ and $d^y$ (Equation 4), which motivates our proposal of using $d^c$ to construct the synthetic twin (Equation 8) and using $d^y$ to validate the trustworthiness.
>
> In Proposition 1, we show that when $d^c$ is minimized to be zero, the ITE error depends on how well we can recover the latent variable $c_i$ using representation learning. Proposition 3 continues to show that the proposed supervised loss function $\mathcal{L}_s$ indeed helps to identify the latent $c_i$ and reduce the ITE error. While Proposition 1 and 2 relate to the training procedure, Proposition 2 shows that we can use $d^y$ to gauge the trustworthiness / identifiability of the individual estimates issued by a trained model - we show that the expected error in ITE can be bounded using $d^y$ and propose a formal procedure to reject untrustworthy estimates.
>
> **Sequential overlap**
>
> We have added a detailed discussion about why SyncTwin does not explicitly require overlap assumption in A 3.2 last paragraph. In summary, SyncTwin requires $d^c$ to be small in order to correctly estimate the ITE; in other words, the target individual should be in or close to the convex hull formed by the controls in the representation space (Equation 2, 4). This condition has a similar spirit to the overlap assumption, which requires (probabilistically) the presence of some control units in the neighbourhood of the target individual in the covariate space.
>
> **Functional linearity assumption**
>
> Your comments about the importance of the functional linearity assumption are exactly correct. As shown in Equation 1, we require the outcomes to be linear in the latent $c_i$ but the irregularly-sampled covariates can relate to the latent $c_i$ in a complex and nonlinear way (which is captured by the neural networks). The functional linearity is central to SyncTwin as it underpins all theoretical results we derive in A 1.
>
> **Minor comments**
>
> We have changed the wording in Section 4 to avoid the impression that SyncTwin can automatically adjust for hidden confounders. We have also added more discussion in A 1.2 to clarify this.

---

### Official Review · AnonReviewer4 · 2020-10-28
**- This paper develops SyncTwin, a method for causal inference with time-series data. The main selling points of the paper are explainability, a small number of donors are used to construct the control, and trustworthiness, the method can identify individuals whose ITE cannot be reliably estimated.**

**Rating:** 4
**Confidence:** 3

**Review:**

- pros
    - The problem is well motivated. Estimating the causal effect with time-series data is a practical and important problem. Transparency is a desirable property.
    - The idea connects works in econometrics to modern ML.
- cons
    - High level points
        - The paper proposes to use the number of donor units as a metric for explainability. It is not clear why this matric is good a priori. It is possible that the network selects an arbitrary set of units to construct the control.
        - For trustworthiness, It is quite confusing how synctwin is supposed to identify individuals whose ITE cannot be reliably estimated. This paper did not provide any formal procedures or guarantees on when identification is possible.
        - Low rank is a key assumption for synthetic control. A discussion on whether this assumption is required for synthetic twins to work would be useful.
        - The experiment section could use more clarification. In the synthetic experiment, the paper compares against a number of benchmarks (including robust synthetic control ) for model performance. However, to show that the method uses fewer contributors, the paper only compared against synthetic control, even though robust synthetic control is a more natural comparator, as it explicitly isolates a small subset of contributors during its data pre-processing.
        - In the real world dataset, 1) while the train. test split is a common practice in ML, it doesn't generalize to inference. Namely, when the estimand is the ATT, we should not use just 1/3 of the data. 2) It is unclear how the ATT is measured.  Is it the average difference at the end of the trial? If so, it may not be a good metric to evaluate the time series model, since it's only the difference in the point estimate.
    - minor details
        - in figure 2, is there a missing arrow between the treatment and the outcome?

---

> ### Author Response · Authors · 2020-11-17
> **Reply to Reviewer 4 (part 2)**
>
>
> **Comparison with Robust Synthetic Control**
>
> We have updated table 7 in A.12 to include the results of Robust Synthetic Control (RSC). RSC includes fewer contributors than Synthetic Control but slightly more than SyncTwin. Note that the donor weights of RSC are not guaranteed to be positive, which is arguably less interpretable than Synthetic Control and SyncTwin (as discussed in Abadie A. 2019). Moreover, we need to use the lasso penalty instead of the standard ridge penalty for RSC to achieve sparsity (see Section 3.4.2 of Amjad M, Shah D, Shen D. 2018).
>
> Compared with the existing works in the Synthetic Control literature, SyncTwin has the unique advantage of allowing the covariates to be observed irregularly over time and have a nonlinear relationship with the outcomes. In addition, SyncTwin also improves the robustness of the original Synthetic Control by avoiding over-match (as shown in the experiments and further discussed in A 3.1) --- the robustness issue of Synthetic Control is widely recognized, and is independently addressed in RSC, MC-NNM, and more.
>
> **Evaluation in the real experiment**
>
> We follow the standard definition of ATT as the conditional expectation $E(\tau_i | a_i = 1)$ (Angrist JD, Imbens GW. 1995). Since the 1/3  testing data is randomly chosen from the entire data set, we can use the average on the testing data to approximate this expectation: $\sum_{i \in \mathcal{D}^{te}_1} \hat{\tau}_{it} / |\mathcal{D}^{te}_1|$. Using the entire data set can potentially lead to a narrower confidence interval around the ATT estimate but it will break the train-test split. In general, whether and when to perform train-test split is still an open research question (see the discussion in Shi C, Blei D, Veitch V. 2019). We choose to perform the split because it is routinely done in existing works and the size of our real data set is big enough to give narrow confidence intervals even after splitting.
>
> We evaluate the treatment effect on the outcome measured one year after treatment initiation (i.e. $\tau_{it}$ where t = 1 year) because this is the metric reported in the trial. Our method can also estimate the effect at multiple time points over shorter or longer horizons, but we cannot validate these estimates using the trial result as they are not reported.
>
> **References**
>
> Munasib A, Rickman DS. Regional economic impacts of the shale gas and tight oil boom: A synthetic control analysis. Regional Science and Urban Economics. 2015 Jan 1;50:1-7.
>
> Abadie A, Diamond A, Hainmueller J. Comparative politics and the synthetic control method. American Journal of Political Science. 2015 Feb;59(2):495-510.
>
> Abadie A. Using synthetic controls: Feasibility, data requirements, and methodological aspects. Journal of Economic Literature. 2019 Aug.
>
> Amjad M, Shah D, Shen D. Robust synthetic control. The Journal of Machine Learning Research. 2018 Jan 1;19(1):802-52.
> Shi C, Blei D, Veitch V. Adapting neural networks for the estimation of treatment effects. InAdvances in Neural Information Processing Systems 2019 (pp. 2507-2517).

---

> ### Author Response · Authors · 2020-11-17
> **Reply to Reviewer 4 (part 1)**
>
> Thank you for the excellent comments and suggestions; we have updated the paper after taking all your comments into account. Please refer to the point-by-point response below and let us know if anything needs further clarification.
>
> **Evaluating explainability**
>
> The number of donor units is a metric routinely examined and reported in applied Synthetic Control studies (e.g. Munasib, A. and Rickman, D.S., 2015, Abadie A, Diamond A, Hainmueller J., 2015). Having few donors means that the domain experts can easily examine each donor – the experts can then decide whether to accept the estimate based on their prior knowledge (e.g. reject the estimate if the algorithm has selected a donor that is very ‘dissimilar’ to the target). On the contrary, it would be time-consuming and impractical to perform such validation if  a large number of donors are involved, therefore making the method less explainable.
>
> With regard to the hypothetical network that always selects a small but arbitrary set of donors, such a network is still explainable – the expert can go through the few donors and find out they are fully arbitrary (and hence decide not to trust the network). Moverover, this network will also have poor mean absolute error in ITE (the metric we use to evaluate estimation accuracy in Section 5.1).
>
> **Trustworthiness**
>
> To further clarify the issue around trustworthiness, we have significantly expanded the discussion in Section 4.1 and added a formal proof in A 1.1 Proposition 2 that $d^y$ can be used to bound the expected error in ITE. The error bound immediately leads to a formal procedure to reject untrustworthy estimates based on the value of $d^y$ and a pre-specified error tolerance threshold $\delta$. This procedure is also detailed in A 1.1.
>
> **Low rank assumption**
>
> We have made it more explicit in the text that SyncTwin also requires the low rank assumption and we have also expanded the discussion in A.4. The low rank assumption is embedded in the data generating assumption (Equation 1), where the rank is the dimensionality of the latent factors c_i and v_i (both assumed to be small).
>
> In practice, the dimensionality of the latent factor c_i is a hyper-parameter. In the sensitivity analysis presented in Appendix A.13, we show that it does not significantly affect estimation accuracy --- this is because we do not explicitly perform matrix completion but only use c_i to find the weights $b_{ij}$ (Equation 8).
>
> We do not impose any low-rank assumption on the covariates X as they cannot be represented as a matrix due to irregular sampling.

---

### Official Review · AnonReviewer1 · 2020-10-28
**Excellent paper on adapting synthetic control-style approaches to time-series data**

**Rating:** 9
**Confidence:** 4

**Review:**

I'm going to keep this review short because I thought this was a really well-motivated and executed paper. The paper builds on the 'synthetic control' approach that is popular in econometrics which uses a weighted sum of the outcomes of individuals in the control group as an estimate of the control potential outcome, which can then be compared to a given treatment outcome to estimate the individual treatment effect. The step is the choice of weights: Sync twin uses the same prediction approach, but constructs the weights for the weighted sum using hidden representations from a recurrent network. This lets them work with irregularly spaced observations & missing covariates in a natural manner.

My only substantive complaint is in the framing of the paper: the introduction describes this temporal setting as more challenging than the static setting, but given that the paper is not about dynamic treatments (which are indeed far harder to deal with), the fact that you see multiple observations for any given individual makes the problem easier not harder... To be clear - this is not bad thing, but rather than saying, "there are many methods for the static setting, but few are able to address temporal confounding" (which isn't really true - these approaches could easily be adapted to the single treatment temporal setting using the appropriate recurrent architectures) - instead emphasize the fact that the static approaches don't take advantage of multiple observations from the same individuals. Multiple observations is a *stronger* assumption than the static setting---albeit a very natural one in this domain that you leverage for trustworthiness, etc.---so it shouldn't be presented as though its a weaker assumption / harder setting...

---

> ### Author Response · Authors · 2020-11-17
> **Reply to Reviewer 1**
>
> Thank you for the excellent comments and suggestions. Based on your comments, we have modified the introduction paragraph to avoid over-claiming that our setting is harder. Having multiple observations over time is a realistic and important setting, but is less studied in the existing literature. Furthermore, we have added a modified version of the CFR-Net (Shalit et al. 2017) as a benchmark in the updated Tables 1, 5, 6. The standard CFR-Net takes a single observation, but we replaced its fully connected encoder with an attentive RNN encoder such that it can handle temporal covariates (SyncTwin uses the same encoder architecture). SyncTwin also performs better than the modified CFR-Net in the experiments.

---

### Official Review · AnonReviewer5 · 2020-11-05
**Some claimed contributions seem to be missing from the final proposed algorithm.**

**Rating:** 4
**Confidence:** 5

**Review:**

Summary:
This paper provides an approach for treatment effect estimation when the observational data is longitudinal (with irregular time stamps) and consists of temporal confounding variables. The proposed method can be categorized under the matching methods, in which, in order to estimate the counterfactual outcomes, a subset of the subjects in the opposite treatment arm (i.e., contributors) is selected and weighted. The proposed method is designed such that it achieves explainability (by identifying a few contributors) and trustworthiness (by checking if the estimated outcome is reliable).

Pros:
- The paper presents a simple, yet reasonable solution to the task at hand.
- The paper is easy to read and understand.
- Figures were very helpful in understanding the paper; especially Figure 2.

Cons:
- There are lots of design choices that are not motivated. Specifically, in section 4.1, the paragraph on Architecture includes too many design choices and heuristics that lack proper explanation and / or intuition. For example, it is unclear why there is need for a new time aware representation $o_{is}$ while we already do have one, i.e., $h_{is}$? Also, $o_{is}$ is mistakenly called a *projection* (in linear algebra terms).
- There are many missing details that are crucial for understanding the algorithm. For example, in Equation (5), it is not stated which parameters are optimized by each loss. I’m guessing $h$, $r$, and $Q$ are found by $\mathcal{L}_s$; and $\[k_0, k_1\]$, $\[s-0, s_1\]$, and $g$ are found by $\mathcal{L}_r$; right? And how are they optimized? Do you alternate between the two objectives? How about $\mathcal{L}_m$ in Equation (6)?
- The authors claim in section 3.1, paragraph 1, lines -6 to -4 that their model does not over-match however, $\mathcal{L}_r$ exactly does that.
-To achieve trustworthiness, the authors state that they minimize $d_i^y$ in Equation (4); however, this term does not show up in any of the objective functions --- see Equations (5) and (6). Also, there should be $M$ values of $d_i^y$; are you aggregating these values into one (e.g., via averaging) or …?


Minor:
- Section 2, paragraph 2, line 3: $t_{is} < 0$ is only for pre-treatment samples. Here, the discussion is on both pre- and post- treatment.
- Section 2, paragraph 5, line -4,-3: $v_i = 0$ and $v_i \neq 0$, did you mean empty set $\emptyset$ instead of $0$?
- Section 3.1, paragraph 1, line 1: Synthetic Control should be title-case.
- Equation (2), the subscript for $\widetilde{c}$ should have been $j$, not $i$.
- Section 4.3, paragraph 1, line -2: $M$ in $\mathcal{L}_M$ should be lower-case.

---

> ### Author Response · Authors · 2020-11-17
> **Reply to Reviewer 5**
>
> Thank you for the excellent comments and suggestions; we have updated the paper after taking all your comments into account. Please refer to the point-by-point response below and let us know if anything needs further clarification.
>
> **Design choices for the Architecture**
>
> We have updated Section 4.1 to include more motivations and references to clarify the design choices. We need to use a  sequence-to-sequence architecture to learn the fixed-sized vector representation $\tilde{c}_i$, but the algorithm is agnostic to the exact choice of the encoder or decoder.
>
> For this reason, we used a standard encoder architecture proposed by Bahdanau D, Cho K, & Bengio Y. (ICLR 2015), which has proven to be successful in many applications. The only difference is that we use the GRU-D network (instead of bi-directional LSTM) because GRU-D can handle irregularly sampled inputs better and it has been successfully applied to a variety of medical time series tasks. The decoder is a standard LSTM network.
>
> With regard to the specific question about time-aware representation, the decoder can only access the aggregated representation $\tilde{c}_i$ but not the ones before aggregation ($h_i$) --- this is because we want $\tilde{c}_i$ to represent the entire sequence on its own (Section 4.1 paragraph 1). Since the timing information could be lost during aggregation, we re-introduce time to inform the decoder when to decode (e.g. the decoder should output values at t = 1, 4, 5, …). Reintroducing timing information during decoding is a standard practice in Seq2Seq models for irregular time-series (e.g. Rubanova Y, Chen RT, Duvenaud DK, NeurIPS 2019; Li, S.C.X. and Marlin, B.M., ICML 2020).
>
> **Training and inference procedure**
>
> The detailed model training/inference procedure is described in Appendix A.8 and we have added the pseudocode to bring more clarity. We optimize the weighted sum of $L_r$ and $L_s$ using end-to-end stochastic gradient descent. As a result, all parameters involved in the encoder and decoder are optimized simultaneously. $L_m$ is only used during inference after the training step is complete. As we highlighted in Appendix A.8, the only trainable parameters in $L_m$ are the weights $b_{ij}$’s (the representations $\tilde{c}_i$ are fixed).
>
> **Avoiding over-match**
>
> We have added a discussion in A.3.1 to illustrate why Synthetic control tends to over-match and SyncTwin does not.
>
> Both SyncTwin and Synthetic Control estimate treatment effects using the weighted combination of control outcomes (Equation 3). Their main difference is in the way to find the weight $b_{ij}$.
>
> Synthetic Control finds $b_{ij}$ by directly minimizing $L_x = ||x_i - \sum_j b_{ij}x_j ||$. Since  $x_i$ contains the observation noise and other random components that do not relate to the outcomes, the weights $b_{ij}$ that minimize $L_x$ tend to over-match, i.e. they captures the irrelevant randomness in $x_i$.
>
> In contrast, SyncTwin finds $b_{ij}$ based on the representations $\tilde{c}_i$ ($L_m$ Equation 8). Since $\tilde{c}_i$ has much lower dimensionality than $\underline{x}_i$, the reconstruction loss $\{L}_r$ encourages $\tilde{c}_i$ to only retain the signal in $\underline{x}_i$ but not the noise. Meanwhile, the supervised loss encourages $\tilde{c}_i$ to only retain the information that predicts the outcomes. As a consequence,  the weights based on $\tilde{c}_i$ are less prone to over-match (as shown in the experiment 5.1).
>
> Moreover, since the relationship between $\tilde{c}_i$ and $x_i$ is nonlinear (as captured by the decoder network), the weights $b_{ij}$ that minimize $L_m$ will generally not minimize the Synthetic Control objective $L_x$, therefore avoiding over-match.
>
> **Trustworthiness and the objective function**
>
> We have significantly expanded Section 4 and added a more formal discussion in A 1.1 Proposition 2  about the trustworthiness. The $d^y$ in Equation 4 is an evaluation metric that indicates whether a trained model can issue reliable estimation for an individual.
>
> We provide justification for the loss functions in A1.1 Proposition 1 and 2, i.e. we show that optimizing our proposed loss functions reduces the ITE error bound.
>
> With regard to the specific question on aggregating multiple values, the || || represents the vector norm in Equation 4; it will aggregate multiple values into a single value.
>
> **References:**
>
> Bahdanau D, Cho K, Bengio Y. Neural machine translation by jointly learning to align and translate. In3rd International Conference on Learning Representations, ICLR 2015 2015 Jan 1.
>
> Rubanova Y, Chen RT, Duvenaud DK. Latent ordinary differential equations for irregularly-sampled time series. InAdvances in Neural Information Processing Systems 2019 (pp. 5320-5330).
>
> Li SC, Marlin BM. Learning from irregularly-sampled time series: A missing data perspective. ICML 2020 Aug 17.

---

> > ### Comment · AnonReviewer5 · 2020-11-24
> > **Thank you for your responses; however, I still think that the paper claims more than it delivers.**
> >
> > I would like to thank the authors for their responses.
> >
> > - Regarding the design choices, I am still not convinced that this architecture really requires as many components for the algorithm to work (e.g., going from $h$ to $c$ and coming back to $o$). More importantly, there is no novelty in the heuristics used here.
> > - Regarding over-match, $L_r$ clearly attempts to reconstruct $X$ in its entirety, with all its noises. Therefore, over-match does happen in SyncTwin as well.
> > - Regarding trustworthiness, the added discussion in A 1.1 Proposition 2 only shows when to reject the estimate. It does not show how such rejection would reduce the ITE error bound though.
> >
> > In summary, I still think that the paper claims more than it delivers. Therefore, I have decided to keep my score.

---

### Official Review · AnonReviewer3 · 2020-11-09
**Interesting concept, unclear use cases, and some technical confusions.**

**Rating:** 3
**Confidence:** 4

**Review:**

## Summary:
The authors develop a method to estimate the effect of a static treatment on outcomes over time under temporal confounding, with an emphasis on making the method transparent. The authors define transparency as (1) the ability to explain the estimate for a given individual as a weighted compination of a small number of other individuals, and (2) trustworthiness meaning the model should be able to identify cases for which it cannot give a reliable estimate due to a violation of one of the main causality assumptions.

## Main comments:
I would significantly increase the rating if the authors address the following comments.

(1) The authors tackle an interesting challenge. I am a bit concerned though that it trivially reduces to ITE estimation with a high dimensional outcome. It seems that the main novelty here is that the authors use a specific architecture that allows for varying lengths of pre-treatment variables, and hence confounders. My concrete questions are (a) suppose for a second that the outcome is measured for a single time period, why does this problem not reduce to the typical ITE estimation? Conditional on all the time varying confounders would simply be equivalent to the typical ITE estimation with a high dimensional set of pre-treatment variables.

(2) The authors only model y(0) for individuals who received the treatment, and define the ITE to be the difference between the observed outcome and \hat{y}(0), and similarly for control individuals. This means that in order to use this model i.e., in order to obtain the ITE estimate for a new test patient, we need to first observe the outcome under t = 1 or t = 0. This makes the model not useful for making treatment decisions: it would only be useful to validated that some chosen treatment choice was good/bad. Can the authors clarify what the intended use for this model would be?

(3) d_i^y is simply a test of whether or not we're able to model the outcomes under t = 0. I do not follow the logical jump from "there is a large estimation error" to there is additional hidden confounding or the data generating model is not right. There are several other things that could cause that (e.g., the model can be well specified, and no hidden confounding but errors arise due to finite sample analysis). To be more concrete, equation 15 in A.1.2 assumes that
(a) the SEM is correctly defined.
(b) Q, U are correct
Either of these might be violated in practice. In fact, they will likely be violated in practice. The flip side is true, one can get a small estimation error and still have hidden confounding. In general, the assumption of no hidden confounders is statistically untestable (meaning it is impossible to design a statistical test that will answer this question). Can the authors highlight what I am missing here?

## Minor comments:
Addressing these would be helpful but not consequential for the rating
(1) The authors make a statement that in general the error in ITE should be bigger than the error in y --> this is not true, for example it does not hold in situations where there is a large bias in opposite directions for estimates in \hat{y}(1), and \hat{y}(0). My comment here assumes the authors are operating in the "usual" setting where we don't observe the treatment and hence the outcome under treatment for a new test case, and need to estimate the ITE by estimating both potential outcomes.

(2) In eq 5 m_{is} is not defined anywhere

(3) The authors don't make a clear case (empirically or in the writing) as to why their chosen approach is transparent. This b vector can still be very high dimensional even with an imposed sparsity. In a sense, all what the authors do with this b vector can be done using a simple kernel approach. Would the authors be able to explicitly state how is their model uniquely equipped for transparency?

(4) It is somewhat odd to have a causal graph where there is no arrow between the treatment and the outcome. After doing a bit of mental gymnastics, I arrived at the conclusion that this graph is assuming do t= 0, which makes sense but is highly unconventional, and confusing. It would be helpful if the authors follow the conventional notation for causal graphs, which does not condition on a particular treatment assignment (e.g., see graphs in publications by Pearl, Eric Tchetgen Tchetgen, Ilya Shpitser, Tyler VanderWeele)

---

> ### Author Response · Authors · 2020-11-17
> **Reply to Reviewer 3 (Part 3)**
>
> **4. (minor questions) Transparency**
> From Equation 3, we can see the estimated ITE depends on a weighted sum of control units (weights are given by vector b).  This means if vector b is highly sparse, the estimate will only depend on only a few individuals. As a result, the domain experts can explain the estimate by going through the short list of contributing individuals. They can further check if the contributing individuals are indeed ‘similar’ to the target individual using domain knowledge.
> This feature is especially important for clinical decision support, where the doctors can compare the patient at hand and the ones who contributed to the ITE estimate. Moreover, we have shown in the real data study that the weights learned by SyncTwin are indeed sparse: on average it selects only 15 contributors from  24,557 individuals (Figure 5).
>
> **5. Other minor questions**
>
> In Equation 8, m_{is} is defined in Section 2 Paragraph 2 (5th line). It is a masking vector indicating which components of x_{is} are observed. All other minor issues (typo, missing arrow in the figure) are addressed in the updated paper.
>
> **References**
>
> Shalit U, Johansson FD, Sontag D. Estimating individual treatment effect: generalization bounds and algorithms. InInternational Conference on Machine Learning 2017 Jul 17 (pp. 3076-3085). PMLR.
>
> Yao L, Li S, Li Y, Huai M, Gao J, Zhang A. Representation learning for treatment effect estimation from observational data. InAdvances in Neural Information Processing Systems 2018 (pp. 2633-2643).
>
> Bica I, Alaa AM, Jordon J, van der Schaar M. Estimating counterfactual treatment outcomes over time through adversarially balanced representations. InInternational Conference on Learning Representations 2019 Sep 25.
>
> Shi C, Blei D, Veitch V. Adapting neural networks for the estimation of treatment effects. InAdvances in Neural Information Processing Systems 2019 (pp. 2507-2517).
>
> Zhang Z, Lan Q, Ding L, Wang Y, Hassanpour N, Greiner R. Reducing Selection Bias in Counterfactual Reasoning for Individual Treatment Effects Estimation. arXiv preprint arXiv:1912.09040. 2019 Dec 19.
>
> Lim B. Forecasting treatment responses over time using recurrent marginal structural networks. InAdvances in Neural Information Processing Systems 2018 (pp. 7483-7493).

---

> ### Author Response · Authors · 2020-11-17
> **Reply to Reviewer 3 (Part 2)**
>
> **2. Estimating new test patients and the intended use case**
>
> SyncTwin is able to estimate the treatment effect on the new patients for whom we have only observed the covariates X (Appendix A.5) and thus can also be used for making treatment decisions. In the paper, we focused on the case of estimating the potential outcome Y(0) for illustrative purposes. To estimate both potential outcomes for a new test patient, we can create two twins: one from the control group and the other from the treated group. The treatment effect can then be estimated as the difference between the two twins. The same error analysis in Appendix A 1.1 also applies in this scenario. The overall error of the estimate is bounded by the sum of the errors in the treated and control outcome estimates.
>
> SyncTwin has important use cases even when it is only used to estimate Y(0) for the treated patients --- it can emulate and extend the clinical trials using observational data (as we show in the real data experiment in Section 5.2). Clinical trials are expensive, time consuming and tend to underrepresent certain populations. Here are a few specific use-cases. (1) We can evaluate the effect of a drug on a wider population than the clinical trials can do (e.g. most clinical trials do not recruit patients above 70 years old for ethical reasons. But we can use SyncTwin to estimate the effect on the older population using observational data). (2) We can identify the unintended therapeutic effect of a drug on a different disease – opening room for drug repurposing (using a drug originally developed for disease A to treat disease B).  (3) We can better identify the heterogeneity of treatment effects across the population using large and representative observational data. (Clinical trials may struggle to identify heterogeneous effects due to small sample sizes)
>
> **3. New theoretical results on trustworthiness and hidden confounding**
>
> To better clarify the trustworthiness, we have added an additional theoretical result (A 1.1 Proposition 2) that shows $d^y$ can be used to bound the expected error in ITE under no hidden confounding. The error bound leads to a formal procedure to reject untrustworthy estimates based on the value of $d^y$ and a pre-specified error tolerance threshold $\delta$. It is true that multiple causes can lead to large $d^y$ including insufficient samples, mis-specified models and hidden confounders (and it has been clarified in the Section 4). However, regardless of the underlying cause, large $d^y$ always indicates untrustworthy estimates.
>
> With regard to hidden confounding, we have clarified in the text that we cannot use $d^y$ to construct a statistical test for no hidden confounder assumption (updated Section 4). However, small $d^y$ implies that the hidden confounders do not significantly influence the outcomes before treatment (even if they exist). The user should choose to believe if these hidden confounders will continue to be insignificant after treatment (e.g. the underlying environment is stable) or they will start to generate a significant impact. In the latter case, no checking procedure based on pre-treatment data will be useful. A detailed discussion can be found in the updated A1.2.

---

> ### Author Response · Authors · 2020-11-17
> **Reply to Reviewer 3 (Part 1)**
>
> Thank you for the excellent comments and suggestions; we have updated the paper after taking all your comments into account. Please refer to the point-by-point response below and let us know if anything needs further clarification.
>
>
> **1. Problem setting and novelty**
>
> Our setting does not reduce to the typical ITE setting as a special case. As illustrated in Figure 1, our setting consists of three sets of variables: the covariates, the outcomes before treatment and the outcomes after treatment --- whereas the typical setting involves the covariates and outcomes after treatment while overlooking the outcomes *before* treatment.
>
> SyncTwin utilizes the problem structure that the outcomes are observed both before and after treatment to inform ITE estimation. As a result, SyncTwin differs from the standard ITE methods in two important aspects conceptually. (1) SyncTwin uncovers a *time-invariant* latent variable c_i that *linearly* relates to the outcomes before and after (Equation 1). The time invariance and the functional linearity allows us to construct synthetic counterfactual outcomes as the weighted averages of other individuals’ outcomes (Equation 3) --- this is in striking contrast with the existing methods which learn a nonlinear function (often a neural net) that maps the covariates directly to the counterfactual outcomes. (2) SyncTwin utilizes the outcomes before treatment to validate the trustworthiness of the ITE estimate at almost no cost --- no architectural change or intensive computation is required --- we only need to compute an evaluation metric $d^y$ (more discussions on this to follow). In contrast, validating trustworthiness is much harder in the standard setting, where it is usually posed as an uncertainty estimation problem and involves change of architecture and intensive computation.
>
> One can conceive two possible ways to incorporate pre-treatment outcomes into the standard ITE methods (1) ignoring these variables and (2) treating them as additional covariates. (1) Throwing away the pre-treatment outcomes is clearly not advisable as we will lose information that is predictive of future outcomes. (2) Treating them as covariates is viable but not good enough: the pre-treatment outcomes are arguably much more closely linked to the post-treatment outcomes than the covariates --- they hence deserve a more “privileged” treatment, e.g. modifying the architecture or loss function to reflect their importance. Without such adjustment, the network will not incorporate our prior belief on the importance of the pre-treatment outcomes, which may lead to worse performance. To illustrate this point, we have added a modified version of the CFR-Net (Shalit et al. 2017) as a benchmark in the updated Tables 1, 5, 6. We replaced the CFR-Net’s fully connected encoder with the same attentive RNN encoder as SyncTwin and used the pre-treatment outcomes as additional covariates. SyncTwin outperforms CFR-Net in all experiments, especially when the data size is smaller (where the prior knowledge on the importance of the pre-treatment outcomes plays a bigger role).
>
> With respect to novelty, SyncTwin’s novelty is multifaceted and extends far beyond the specific architecture choice. (1) Novelty to the machine learning ITE literature. SyncTwin is inspired by the Synthetic Control method in Econometrics. It is an alternative to the two mainstream approaches in ML-based ITE estimation, i.e. balancing representations (e.g. Shalit et al. 2017, Yao et al. 2018, Bica et al. 2020) and propensity score (e.g. Shi et al. 2019, Hassanpour et al. 2019, Lim et al. 2018) -- SyncTwin does not balance the treated and control groups by penalizing their distributional distance or by propensity weighting -- instead it creates a synthetic twin for each individual as the weighted averages of other individuals. (2) Novelty to the Synthetic Control literature. Compared with the existing works in the Synthetic Control literature, SyncTwin has the unique advantage of allowing the covariates to be observed irregularly over time and have a nonlinear relationship with the outcomes. In addition, SyncTwin also improves the robustness of the original Synthetic Control by avoiding over-match (as shown in the experiments and further discussed in A 3.1)

---

### Author Response · Authors · 2020-11-17
**Revised paper**

We thank all reviewers for reviewing and commenting on our manuscript.  All of your inputs have been valuable, and have helped to improve the manuscript.

We have updated the manuscript to incorporate the constructive feedback.  We have indicated our revisions in the revised manuscript by blue text.  Please see individual responses to reviewers for revision details.

Here we provide a brief summary of the major revisions to the manuscript:
1) We have significantly expanded Section 4 to include more rigorous expositions.
2) We have expanded Section 4.1 to include more references and motivation.
3) We have included a new benchmark in the synthetic experiment in Section 5.
3) We have added a new proof in Appendix 1 and more detailed discussions around trustworthiness and hidden confounding.
4) In Appendix 3.2, we added a paragraph to explain "Why does Synthetic Control tend to over-match?"
5) In Appendix 3.3, we added a paragraph to explain "Why does SyncTwin not explicitly require overlap?"
6) In Appendix 8, we added the pseudocode for training and inference procedure.

Please let us know if further clarifications are needed.

---

### Decision · Program_Chairs · 2021-01-07
**Final Decision**

**Decision:**

Reject

**Comment:**


The paper developed a method that estimates treatment effect with
longitudinal observational data under temporal confounding. It extends
the idea of the synthetic control method and offers flexibility and
ease of estimation. However, some major concerns remain after the
discussion among the reviewers. In particular, the proposed method
lacks a clear use case. Moreover, some arguments around
``trustworthiness`` (detecting unreliable ITE estimates) and ``avoid
over-matching`` need to be refined. The error bound for
``trustworthiness`` can not detect hidden confounding. For overlap
issues, the rejection of units with larger error could be overly
conservative because the error bound may often be too loose. Regarding
``avoid over-matching``, while SyncTwin uses a low-dimensional
representation as opposed to the whole x vector for matching, it is
unclear whether SyncTwin can avoid over-match. It is possible that
using low-dimensional representation makes it easier to find a match
in the data and may still over-match. Finally the paper would benefit
from proper causal identification results.